

**Title:** Perturbation increases source-dependent organic matter degradation rates in estuarine
sediments.
**Author information**
Guangnan Wu[1], Klaas G.J. Nierop[2], Bingjie Yang[3], Stefan Schouten[3], Gert-Jan Reichart[1, 2], Peter
Kraal[1]
[1] Royal Netherlands Institute for Sea Research, Department of Ocean Systems, Landsdiep 4, 1797
SZ 't Horntje, The Netherlands
[2] Utrecht University, Faculty of Geosciences, Princetonlaan 8a, 3584 CB Utrecht, The Netherlands
[3] Royal Netherlands Institute for Sea Research, Department of Marine Microbiology &
Biogeochemistry, Landsdiep 4, 1797 SZ 't Horntje, The Netherlands
**Corresponding author**
Guangnan Wu (guangnan.wu@nioz.nl)



**Abstract**
Despite a relatively small surface area on Earth, estuaries play a disproportionally important role in
the global carbon cycle due to their relatively high primary production and rapid organic carbon
processing. Estuarine sediments are highly efficient in preserving organic carbon and thus often rich
in organic matter (OM), highlighting them as important reservoirs of global blue carbon. Currently,
these habitats are facing intensified human disturbance, one of which is sediment dredging. To
understand estuarine carbon dynamics and the impact of perturbations, insights into sediment OM
sources, composition, and degradability is required. We characterized the sediment OM properties
and oxidation rates in one of the world's largest ports, the Port of Rotterdam, located in a major
European estuary. Using a combination of OM source proxies and end-member modeling analysis,
we quantified the contributions of marine (10–65%), riverine (10–60%), and terrestrial (10–65%) OM
inputs across the investigated transect, with salinity ranging from 32 (marine) to almost 0 (riverine).
Incubating intact sediment cores from two contrasting sites (marine versus riverine) suggested that
OM was more reactive in marine sediment than riverine sediment. Exposing wet bulk surface
sediment to atmospheric oxygen in a bottle incubation experiment showed a 2.8–7.4 times increase of
OM degradation rates, while the impact of OM source and composition maintained the observed
differences in rates between sites. This shows that sediment perturbation and the reintroduction of
oxygen can substantially boost OM degradation. By combining detailed quantitative characterization
of estuarine OM properties with degradation rates under different environmental conditions, our
results further our understanding of the factors that govern OM degradation rates in (perturbed)
estuarine systems. Ultimately, this contributes to constraining the impact of human perturbation on
OM cycling in estuaries and its role in the carbon cycle.

**Graphical abstract**

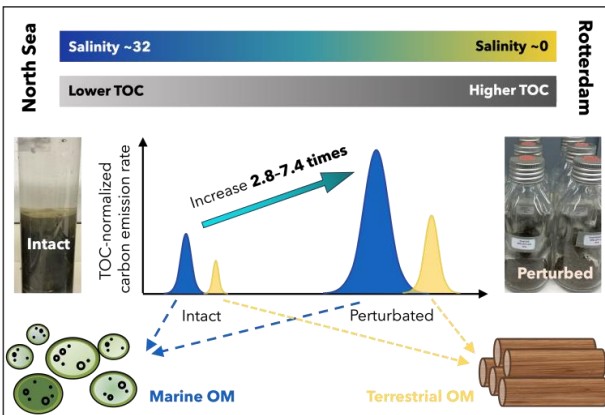







## 1. Introduction

Estuaries are highly dynamic aquatic systems that are influenced by simultaneous marine, riverine, and terrestrial inputs. In this transition zone, strong and variable gradients exist in hydrodynamic and sediment properties, resulting in dynamic and complex cycles of key elements such as carbon through coupled physical, chemical, and biological processes (Barbier et al., 2011; Dürr et al., 2011; Laruelle et al., 2010). Despite representing only 0.03% of the surface area of marine systems, estuaries are estimated to release approximately 0.25 Pg carbon annually into atmosphere on a global scale, which is equivalent to 17% of the air-water $CO_2$ gas exchange of the entire open ocean (Bauer et al., 2013; Li et al., 2023). Additionally, estuarine sediments store large amounts of organic carbon (Macreadie et al., 2019; McLeod et al., 2011); due to high productivity and high sedimentation rates, carbon burial rates in estuaries are up to one order of magnitude higher than forest soils and three orders of magnitude higher than in open ocean sediments (Kuwae et al., 2016). Their disproportionally large importance in the global carbon cycle highlights the need to improve our understanding of carbon dynamics in estuarine systems.

Organic matter (OM), a fundamental component of sediment, plays a key role in sediment carbon fluxes and sequestration. The degradation of OM contributes to the release of carbon dioxide ($CO_2$) and methane ($CH_4$). It is a dynamic process that proceeds through a series of enzymatic reactions involving different organisms, oxidants, and intermediate compounds. Studies have pointed out the importance of OM characteristics in influencing the rate and extent of OM degradation (Burd et al., 2016; Burdige, 2007; LaRowe and Van Cappellen, 2011). For instance, extensively degraded OM and biopolymers such as cellulose and lignin are less susceptible to degradation than freshly produced nitrogenous compounds (Arndt et al., 2013). Estuarine systems have diverse terrestrial and aquatic OM sources, which exhibit varying degrees of degradability (Canuel and Hardison, 2016). Moreover, the interactions between OM and other components (organic or inorganic) during transportation, deposition, and mineralization can alter OM characteristics. Processes such as condensation, (geo)polymerization and mineral association increase the resistance to OM degradation, thereby promoting OM preservation (Wakeham and Canuel, 2006).

Sediment OM degradation is also influenced by ambient environmental conditions (Arndt et al., 2013; Burd et al., 2016; Burdige, 2007; LaRowe and Van Cappellen, 2011). The degradation pathway follows the sequential utilization of the terminal electron acceptors (TEAs), typically in the order of $O_2$, $NO_3^-/NO_2^-$, Mn (IV), Fe (III) and $SO_4^{2-}$, with a progressive decrease in energy yield down the redox ladder. The availability of these TEAs is greatly influenced by the depositional conditions. Estuaries are highly dynamic systems where strong and shifting salinity (i.e. sulfate) gradients exist. This can lead to a strong spatial variability in OM degradation pathways and carbon dynamics, particularly for $CH_4$ (Cao et al., 2021). Moreover, compilation of field data reveals that organic carbon burial efficiency varies substantially in space because the availability and exposure time of TEAs are influenced by environmental factors such as sedimentation rate (Arndt et al., 2013; Freitas et al., 2021). Estuaries are often characterized by relatively high sedimentation rates, with supply of riverine material that





settles under low flow velocities in deltas and estuaries as well as large inputs of (re)suspended
marine matter from the coastal zone (ref). Oxygen transport into sediment is sufficiently low relative to
the flux of reactive organic carbon to sediments to maintain very shallow oxygen penetrations depths,
on the scale of micro- to millimeters (Burdige, 2012). By notably reintroducing $O_2$ to previously buried
OM in oxygen-deficient environment, both naturally and anthropogenically induced sediment
disturbance can change sediment redox chemistry and thereby have a profound effect on OM
degradation pathways and burial efficiency (Aller, 1994).

Although estuaries have been widely studied from an ecological perspective, large variation in OM
properties and cycling processes within and across estuarine systems contributes to the uncertainty in
quantifying their significance in the global carbon cycle. This uncertainty is partially due to the highly
diverse OM sources and properties in estuarine systems. Many studies of estuarine OM sources use
bulk proxies such as the weight ratio of total organic carbon to total nitrogen (C/N ratio) and their
stable isotope ratios ($\delta^{13}C_{org}$ and $\delta^{15}N$; (Canuel and Hardison, 2016; Carneiro et al., 2021; Cloern et
al., 2002; Middelburg and Nieuwenhuize, 1998). In other studies, OM sources have been investigated
by identifying biomarker compounds that are associated with specific sources and transformation
processes. For example, the branched and isoprenoid tetraether (BIT) index, based on the relative
abundance of terrestrially and/or freshwater derived branched glycerol dialkyl glycerol tetraether
(GDGT) versus marine derived isoprenoid GDGT crenarchaeol, was adopted to quantify the relative
contribution of terrestrial OM in sediments (Herfort et al., 2006; Hopmans et al., 2004; Smith et al.,
2010; Strong et al., 2012). Some studies focused on macromolecular organic matter (MOM)
composition in sediments to identify OM sources (Kaal et al., 2020; Nierop et al., 2017). Lignin, an
important constituent of vascular plant MOM, has proved to be a useful tracer of vascular plant inputs
to estuarine/coastal margin sediment (Bianchi and Bauer, 2012; Buurman et al., 2006; Fabbri et al.,
2005; Hedges and Oades, 1997; Kaal, 2019). Furthermore, the relationship between OM source and
degradability can be intricate, which inhibits our quantitative understanding of estuarine OM
degradation.

Understanding the processing of OM within estuaries takes on further importance because many
estuarine systems are intensively altered by human activities (Arndt et al., 2013; Heckbert et al.,
2012; Holligan and Reiners, 1992). Dredging is a common sediment management practice in many
coastal regions and rivers worldwide. More than 600 million $m^3$ of dredged material is generated
annually just in Western Europe, China, and the USA (Amar et al., 2021). These anthropogenic
perturbations expose buried sediment to an oxygenated environment, which is energetically favorable
for OM degradation (LaRowe et al., 2020). The active sediment reworking on the Amazon shelf was
reported to stimulate mineralization and decreased the sediment organic carbon content (Aller et al.,
1996). Considering the massive amount of material being dredged, recent studies have suggested to
explore the possibilities of reusing dredging sediment as construction materials (Brils et al., 2014).
However, one of the great unknowns lies in the fate of the large amount of organic carbon stored in
these sediments during dredging, drying, processing, and further use. Given that dredging activities



continue to increase driven by the increasing societal and economic needs (van de Velde et al.,
2018), it is of great importance to understand to what extent anthropogenic sediment perturbations
affect OM processing in and carbon emissions from estuarine sediments.
In this study, we investigate the spatial variability in OM content and properties and relationships
between OM source, composition, and degradability along a salinity gradient in the profoundly
disturbed Port of Rotterdam estuarine environment of the Rhine-Meuse delta system. Given the
frequent dredging activities in our study area, which hosts a globally major port, we aim to understand
the impact of sediment dredging and its potential land applications on carbon dynamics. We used a
combination of bulk OM proxies, BIT index, macromolecular organic matter (MOM) composition
analysis, as well as end-member modelling to understand OM sources and composition. Furthermore,
organic matter degradation rates were estimated both in undisturbed sediment cores and in bottles
incubation with wet sediment under atmospheric conditions, the latter as representative for dredged
sediment. Our show that variability in OM sources and subsequently molecular properties, as well as
perturbation (i.e. introduction of oxygen), have important effects on OM degradation rates. We show
that in addition to content, the properties of OM influences carbon emissions from estuarine sediment
and the carbon footprint of anthropogenic perturbations.

**2. Materials and methods**
**2.1. Study area and sample collection**
Our study area is located in the northern part of the Rhine-Meuse estuary (Fig. 1), spanning from
Rotterdam city to the Maasmond. This area representing a transitional environment from riverine to
marine is heavily urbanized and hosts one of the world's largest ports, the Port of Rotterdam (PoR).
Every year, large amounts of sediment are deposited in the harbor from both rivers as well as the
North Sea (Kirichek and Rutgers, 2020). The water channel maintenance and harbor expansion lead
to an increasing need of sediment dredging. Currently, over 10 million $m^3$ of dredged materials are
relocated to the shallow North Sea, while around 1.5 million $m^3$ are being stored as contaminated
sediment in a holding basin in the PoR area (Kirichek and Rutgers, 2020).
We collected bulk sediments from 49 locations throughout the study area (Fig. 1) in the summer of
2021. Sediments down to ~50 cm depth were collected using a gravity corer (⌀9 cm). Once on deck,
materials in the corer were emptied into 5-L polypropylene buckets that were closed and stored in the
fridge at 4 °C. These samples, later referred as bulk sediments, were further processed within a week
after collection at the Royal Netherlands Institute for Sea Research (NIOZ) on Texel, the Netherlands.
In addition to bulk sediments, intact sediment cores were collected in summer 2022 upon revisiting
two contrasting sites (referred as 'super sites' in Fig. 1) in the marine (site 115, salinity 28.7) and
riverine (site 21A, salinity 5.1) realm of the PoR area. The intact sediment cores were immediately
cooled, transported back to the NIOZ and used in whole-core incubation experiments (see section
2.5) within 5 hours after collection.




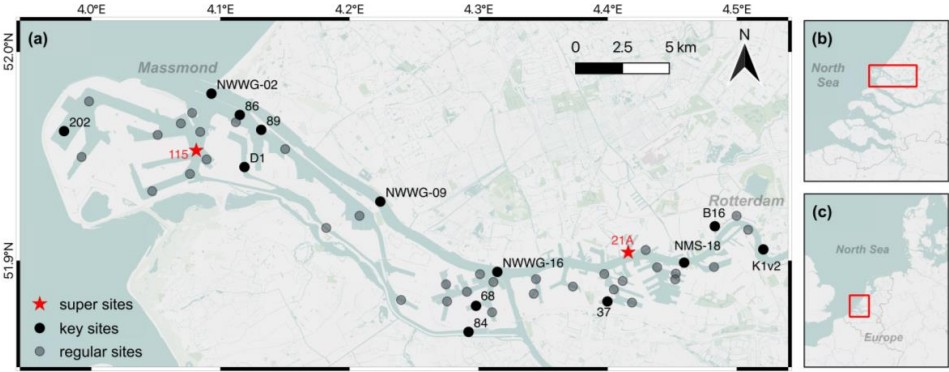

**Fig. 1.** (a) The investigated study area and sampling sites. Sediments from all 49 sites were subjected
to bulk analysis as detailed in section 2.2. Sediments from 13 key sites were used for lipid and MOM
analysis as detailed in section 2.3 and 2.4, respectively. Sediment cores from two super sites were
used in a whole-core incubation experiment as detailed in section 2.5. (b) The location of investigated
study area in the Rhine–Meuse estuary. (c) The location of Rhine-Meuse estuary in Western Europe.
Map created using QGIS software. Basemap courtesy of Mapbox.

## 2.2. Sample processing and analysis

Bulk sediments were thoroughly mixed using a spatula in the buckets. Approximately 40 mL of wet
sediment were transferred into 50-mL polypropylene centrifuge tubes (Falcon) and centrifuged at
3000 rpm for 20 min (Hermle Z 446). In a $N_2$-purged glove bag, the porewater was immediately
filtered through a 0.45-μm nylon syringe filter (MDI). Salinity was estimated by comparing the
porewater sodium (Na) concentration to the average seawater sodium concentration and salinity in
the North Sea (IJsseldijk et al., 2015; Steele et al., 2010). For Na analysis, the porewater was diluted
around 900 times in 1 M double-distilled $HNO_3$ and analyzed by inductively coupled plasma mass
spectrometry (ICP-MS, Thermo Scientific, Element 2).

The centrifuge tubes with wet sediment residues after centrifugation were purged with $N_2$ and stored
at −20 ºC in $N_2$-purged, gas-tight Al-laminate bags to prevent oxidation. To prepare for subsampling,
the sediment residues were thawed overnight in a $N_2$-purged glove bag (Coy Laboratories) and
subsequently homogenized. One portion of wet sediment residue (~1 g) was mixed with 50 mL of 3 g
$L^{-1}$ sodium pyrophosphate solution and gently shaken to disaggregate particles. Particle size
distribution was determined using a Coulter laser particle sizer (Beckman Coulter), from which
percentages of clay (0–2 μm), silt (2–63 μm), sand (63–2000 μm) and the median particle size (D50)
were calculated.

Approximately 10 g wet sediment residue was freeze-dried (Hetosicc freeze dryer) for 72 h and
manually ground with an agate pestle and mortar, and further subsampled for carbon and nitrogen
(CN) analysis. One subsample of the freeze-dried sediment (~10 mg) was directly used for measuring



total nitrogen (TN) and stable nitrogen isotope composition (expressed as $\delta^{15}$N, relative to
atmospheric nitrogen) by a CN elementary analyzer (Thermo Scientific, FLASH 2000) coupled to a
Delta V Advantage isotope ratio mass spectrometer (Thermo Scientific). Another freeze-dried
subsample (~0.5 g), firstly treated with 1 M HCl to remove carbonates, was used for measuring total
organic carbon (TOC) and stable carbon isotope composition (expressed as $\delta^{13}$C$_{org}$, relative to Vienna
Pee Dee Belemnite). Certified laboratory standards (acetanilide, urea, and casein) were used for
calibration with each sample. Precision and accuracy for standards and triplicate samples were
±0.3‰ for $\delta^{13}$C$_{org}$ and $\delta^{15}$N, and the relative standard deviation (RSD; standard deviation/mean) was
<10% for TOC and TN.

**2.3. Lipid extraction and analysis**
Freeze-dried and homogenized sediments (2–10 g) from 13 key locations (Fig. 1) were ultrasonically
extracted with dichloromethane (DCM):methanol (2:1, v:v) five times. For each sample, extracts
obtained from the five steps were combined. The total extract was separated over an Al$_2$O$_3$ column
into an apolar, neutral and polar fraction using hexane:DCM (9:1, v:v), hexane:DCM (1:1, v:v) and
DCM:methanol (1:1, v:v), respectively. The polar fractions containing glycerol dialkyl glycerol
tetraethers (GDGTs) were dried under N$_2$, dissolved in hexane:propanol (99:1, v:v), and filtered using
a 0.45 µm PTFE filter. This fraction was subsequently analyzed with an ultra-high performance liquid
chromatography mass spectrometry (UHPLC-MS) on an Agilent 1260 Infinity HPLC coupled to an
Agilent 613MSD according to (Hopmans et al., 2016). The isoprenoid and branched GDGTs were
detected by scanning for their [M+H]$^+$ ions. The BIT index was calculated according to (Hopmans et
al., 2004).

**2.4. Macromolecular organic matter (MOM) isolation and analysis**
The sediment residues after lipids extraction were dried under N$_2$. To isolate MOM, dried sediment
residue (2–3 g) was transferred into 50-mL centrifuge tubes and decalcified with 30 mL 1 M HCl for 4
h, later rinsed twice with 25 mL milli-Q water (18 MΩ). After centrifugation and decanting the
supernatant, 15 mL 40% HF (analytical grade, Merck) was added and shaken for 2 h at 100 rpm. The
solution was diluted with milli-Q water to 50 mL and left standing overnight, after which the solution
was decanted. A volume of 15 mL 30% HCl was added and subsequently diluted with milli-Q water to
50 mL. After shaking for 1 h and centrifugation, the solution was decanted, and the residues were
washed with milli-Q water three times to neutralize pH and subsequently freeze-dried. Samples were
desulfurized using activated copper pellets in DCM. Suspensions were stirred overnight after which
the copper pellets and DCM were removed, and the MOM was air-dried prior to the analysis.

The analysis of MOM was conducted at Utrecht University using the pyrolysis-gas chromatograph-
mass spectrometry method previously described in (Nierop et al., 2017). In short, the isolated MOM
was pyrolyzed on a Horizon Instruments Curie-Point pyrolysis unit. The pyrolysis unit was connected
to a Carlo Erba GC8060 gas chromatograph and the products were separated by a fused silica
column (CP-Sil5, 25 m, 0.32 mm i.d.) coated with CP-Sil5 (film thickness 0.40 µm). The column was



coupled to a Fisons MD800 mass spectrometer. Pyrolysis products were identified using a NIST
library or by interpretation of the spectra, by their retention times and/or by comparison with literature
data. Quantification was performed according to (Nierop et al., 2017).

**2.5. Whole-core sediment incubation**
Triplicate intact sediment cores collected from sites 115 and 21A were used for whole-core incubation.
Prior to incubation, cores were carefully manipulated to have ~15 cm of undisturbed top sediment with
~20 cm of overlying water. After confirming that the sediment surface was not disturbed, an oxygen
sensor spot (Presens) was attached to the inner wall of the core tube (5 cm from the top) to monitor
$O_2$ in the overlying water. The cores, capped at the bottom and open at the top, were submerged in
bottom water from the corresponding site in an incubation tank. Stirrers were placed in each core to
mix the overlying water (at ~1 rpm) and the cores were left open overnight to equilibrate. The water in
the tank was kept fully oxygenated by sparging with air using an aquarium pump. Temperature in the
room was maintained at the measured site bottom water temperature (19 °C). At the start of the
incubation, the cores were capped with gas-tight lids with an outlet to sample bottom water in core
and an inlet to replace sampled volume with site water from a 20-L reservoir. Over the course of an
eight-hour incubation period, 30 mL of bottom water were extracted at pre-determined time intervals
of 0, 1.5, 3.5, 5, 6.5, and 8 h. The dissolved $O_2$ concentration in the overlying water in each core was
measured every five minutes using the sensor spots and a Presens OXY-4 SMA meter with fiber optic
cables, operated using Presens Measurement Studio 2. Immediately after sampling, the water
samples were filtered using 0.45-µm nylon syringe filters for dissolved inorganic carbon (DIC) and
dissolved inorganic nitrogen (DIN: $NH_4^+$, $NO_3^-$, $NO_2^-$) analysis, while an unfiltered subsample was
retained for methane ($CH_4$) analysis.

The DIC samples were diluted 10 times in $N_2$-purged 25 g $L^{-1}$ sodium chloride solution without
headspace and analyzed within 24 hours by a continuous flow analyzer (QuAAtro, Seal Analytical).
The DIN samples were stored at −20 °C and later analyzed by a continuous flow analyzer (TRAACS
800+). For $CH_4$, 12 mL of bottom water was directly transferred into a 12 mL Exetainer vial (Labco),
immediately poisoned with ~0.25 mL of saturated zinc chloride solution and capped with a butyl
rubber stopper ensuring no headspace was present. Dissolved $CH_4$ concentration was determined
using a headspace technique (Magen et al., 2014). Prior to the measurement, 1 mL of $N_2$ headspace
was injected through the stopper in each Exetainer vial while a needle allowed the equivalent volume
of sample to escape, after which the samples were equilibrated for a week. Headspace $CH_4$
concentrations were then measured by a gas chromatograph (Thermo Scientific FOCUS GC)
equipped with a HayeSep Q Packed GC Column and a flame ionization detector. A calibrated curve
was made using a certified 1000 ppm $CH_4$ standard (Scott Specialty Gases Netherlands B.V.). From
the measured $CH_4$ concentration in the headspace, the total dissolved $CH_4$ in the bottom water was
calculated using the equations in (Magen et al., 2014) with the Bunsen coefficient (Yamamoto et al.,
1976). Benthic fluxes of DIC and $CH_4$ were calculated using the concentration changes of solutes in



the bottom water of closed cores during the incubation period, as determined by linear regression
analysis of the individual time series.

**2.6. Subaerial incubation of dredged sediment**
The subaerial incubation experiments were conducted in triplicate for six sediments (115, 21A, 86,
B16, NWWG-02 and K1v2). Freeze-dried and homogenized sediment (~10 g) was transferred into a
330-mL borosilicate glass bottle, leading to a thin layer (less than 5 mm) of sediment. The moisture
level of sediment was adjusted with artificial rainwater (composition detailed in Table S1 in
Supplementary Information (SI)) to ensure a water-filled pore space at 60% according to (Fairbairn et
al., 2023). The sediment was incubated in the dark at room temperature (20 ℃). The $CO_2$ emission
rate was measured on day 2, 6, 9, 16, 23, 30 and 37. On the day of measurement, bottles were
sealed with rubber stoppers tightened with aluminum crimp caps for approximately 3 hours. We
measured the $CO_2$ concentrations in the headspace immediately after the bottles were capped and
approximately 3 hours later. The $CO_2$ accumulation in the headspace of each bottle during these 3
hours was used to calculate a $CO_2$ emission rate. For the rest of the time, bottles were kept open to
the atmosphere. The moisture level was maintained once a week and varied by less than 10% from
the target value.

The $CO_2$ measurement for the subaerial incubation was conducted by withdrawing a volume of 150
μL headspace gas using a 250-μL glass, gas-tight syringe (Hamilton). The headspace sample was
immediately injected into a gas chromatograph (GC, Agilent, 8890 GC system) equipped with a
Jetanizer and a flame ionization detector. Gases were carried by helium and separated by a
Carboxen-1010 PLOT analytical column (Sigma-Aldrich). Calibration was conducted by using certified
reference $CO_2$ gas (Scott specialty gases, Air Liquide, Eindhoven, The Netherlands).

To determine the percentage of degraded TOC over time, we firstly calculated the cumulative amount
of $CO_2$ emission and then normalized it to the total amount of organic carbon in the incubated
sediments, calculated from the dry sediment mass and its TOC content. The cumulative $CO_2$ emission
was obtained by integrating the $CO_2$ emission rate over time. For days when $CO_2$ emission rates were
not measured, the rates were estimated using spline interpolation. The integration and normalization
were performed using the 'AUC' (area under curve) function in RStudio.

**2.7. End-member modelling of OM sources**
The contribution of three major OM end-members (marine, riverine, and terrestrial OM) to the
sediment was quantified based on $\delta^{13}C_{org}$ and C/N ratio using a Bayesian mixing model, MixSIAR
(Stock et al., 2018). Anthropogenic OM such as petroleum and coal products were not considered as
they typically have a much higher C/N ratio (Tumuluru et al., 2012) compared to our samples (mostly
<20), thus suggesting a limited contribution. Input from industrial and chemical waste is considered
being minimal because >90% of sediment is regarded as clean/safe with organic contaminants below
their national intervention values (Kirichek and Rutgers, 2020). We did not include sewage OM and





agricultural wastes as separate end-members due to their high variability in $\delta^{13}C_{org}$ (−28‰–−23‰;
(Shao et al., 2019)) and C/N ratio (Chow et al., 2020; Puyuelo et al., 2011; Szulc et al., 2021), and the
values are largely overlaps with those of the considered three end-members. The model incorporates
the common ranges of three OM end-members in coastal environment (Table 1) and employs Markov
Chain Monte Carlo (MCMC) simulation to sample from the posterior distribution. The distribution
provides estimates of the mean contribution with standard deviation. The model was run in RStudio
with package "MixSIAR" integrated into the JAGS program.

**Table 1.** Mean values and standard deviations of $\delta^{13}C_{org}$ and C/N ratio of three OM end-members
used in the MixSIAR analysis. Values from literature (Bianchi and Bauer, 2012; Finlay and Kendall,
2007; Lamb et al., 2006).

| End-member | Typical OM | $\delta^{13}C_{org}$ (‰) | C/N |
|---|---|---|---|
| Marine OM end-member | Marine POC, algae, bacteria | −20±4 | 7±3 |
| Riverine OM end-member | Freshwater POC, algae, bacteria | −29±4 | 7±3 |
| Terrestrial OM end-member | Vegetation, soil OM, bacteria | −26.5±5.5 | 30±18 |


**3. Results**
**3.1 Bulk geochemical feature of sediments**
The PoR sediments were mostly (42 out of 49 samples) silt-rich with D50 smaller than 20 μm. A
salinity gradient was observed in the study area increasing from approximately 0 at the most eastern
part (Rotterdam city) to approximately 32 at the river mouth in the west. We observed a decrease in
TOC content with increasing salinity (Fig. 2a). The silt-rich sediments generally contained more than
2.5 wt.% TOC, with significantly lower TOC contents in the sand-rich sediments ($p < 0.01$, Student's $t$-
test). The weight ratio of C/N was between 5 and 13 for most samples (45 out of 49), and the
corresponding $\delta^{13}C_{org}$ was in the range of -29‰ to -23‰ (Fig. 2b). Despite a weak correlation
between C/N ratio and $\delta^{13}C_{org}$ ($R = −0.38$, Pearson), both properties showed (moderately) strong
trends against salinity (C/N ratio: $R = −0.66$; $\delta^{13}C_{org}$: $R = 0.68$, Pearson; Fig. 2b).

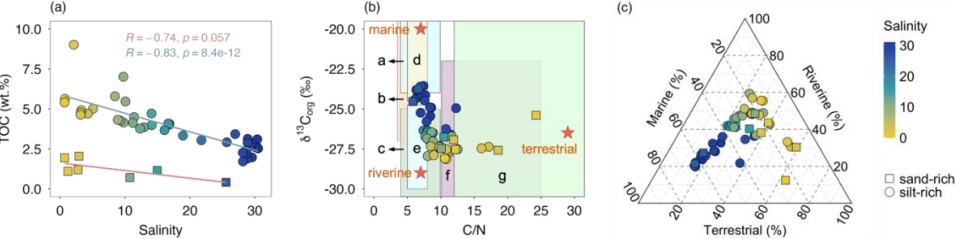


**Fig. 2.** Bulk geochemical properties of 49 sediment samples from the PoR. (a) TOC vs. salinity for
both silt-rich (D50 < 20 μm) and sand-rich (D50 > 50 μm) sediments. (b) $\delta^{13}C_{org}$ and the weight ratio of
C/N in sediments along salinity gradient in contrast to the typical $\delta^{13}C_{org}$ and C/N ranges for OM from
coastal sediments in literature (Bianchi and Bauer, 2012; Finlay and Kendall, 2007; Lamb et al.,
2006): **a** marine POC, **b** bacteria, **c** freshwater POC, **d** marine algae, **e** freshwater algae, **f** soil OM, **g**



C$_3$ terrestrial plants. Asteroid signs represent the mean values of three OM sources used in end-
member analysis. (c) The contribution (%) of marine, riverine and terrestrial OM using a mixing model.
The standard deviation (10–25%) is provided in the Supplementary Information (SI, Table S1).

**3.2. Flash pyrolysis products of MOM**
Pyrolysis of isolated MOM produced hundreds of pyrolysis compounds. The identified pyrolysis
products are listed in Supplementary Information (Table S2). They were divided into nine groups
based on the chemical characteristics, following the approach detailed in Nierop et al. (2017). Here in
Fig. 3, we present the relative abundance of six MOM pyrolysate groups along the salinity gradient,
including $n$-alkenes/alkanes, guaiacols, N-compounds, phenols, polysaccharide-derived products, and
syringols. The other three groups: phytadienes and pris-1-ene were only minor constituents (relative
abundance < 5%), and aromatics showed a negligible correlation with salinity (−0.1 < $R$ < 0.1,
Pearson; Fig. S1). With increasing salinity, we observed an increase in the relative abundance of $n$-
alkenes/alkanes and N-compounds, while guaiacols, phenols, polysaccharides, and syringols
decreased. The correlations were generally moderate or weak, as suggested by the magnitude of the
correlation coefficient (−0.6 < $R$ < 0.6, Pearson). Additionally, the correlation coefficients between the
identified MOM pyrolysate groups and other bulk sediment properties (i.e. D50, C/N, $\delta^{13}C_{org}$) were
also weak (see SI, Fig. S2).

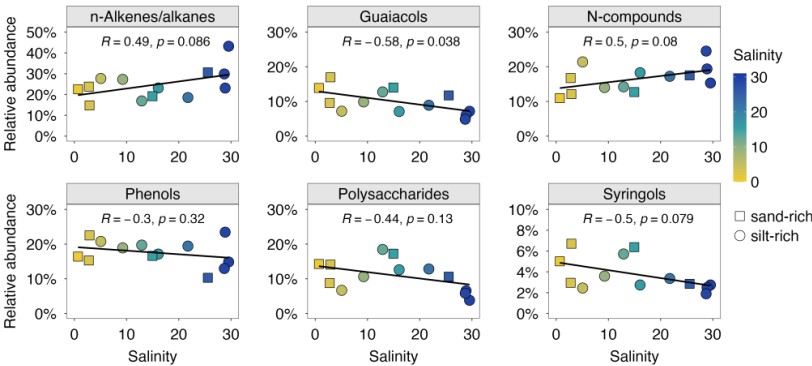


**Fig. 3.** The relative abundance of six groups of MOM pyrolysis products. Pearson correlation
coefficient ($R$) measures the strength of the linear relationship between grouped pyrolysates and
salinity.

**3.3. BIT index**
Crenarchaeol and branched GDGTs were detected in sediments from all 13 investigated sites. The
calculated BIT index ranged between 0.43 and 0.92 (Fig. 4a). A strong negative correlation was
observed between BIT index and salinity ($R$ = −0.88, Pearson) and between BIT index and $\delta^{13}C_{org}$ ($R$
= −0.83, Pearson). In contrast, the correlation with MOM pyrolysis products were in general weak or
moderate (−0.6 < $R$ < 0.6, Pearson; Fig. S2), except for guaiacols and N-compounds (Fig. 4b & 4c).



Additionally, we did not observe significant difference between sand-rich and silt-rich sediments in BIT
index values ($p > 0.5$, Student's $t$-test).

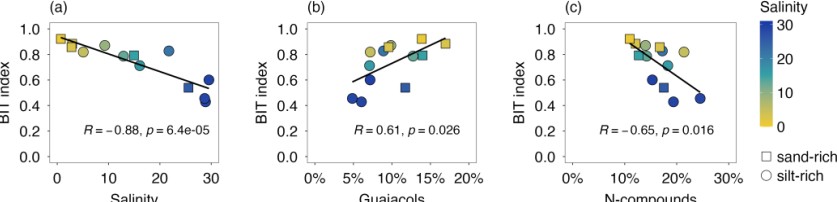


**Fig. 4.** The BIT index of 13 sediments against (a) salinity, (b) relative abundance of guaiacols, (c)
relative abundance of N-compounds.

**3.4. Benthic fluxes on intact sediment cores**
During the whole-core incubation, the $O_2$ concentration in the overlying decreased linearly from
around 90% to 60% air-saturation for both the high salinity location (115, salinity 28.7, later referred as
'marine' location) and the low salinity location (21A, salinity 5.1, later referred as 'riverine' location; SI
Fig. S2). At the same time, concentrations of DIC and $CH_4$ in the overlying water increased linearly
with time (Fig. S2). Benthic $O_2$ consumption rates were very similar at the two contrasting locations,
around 30 mmol m$^{-2}$ d$^{-1}$ (Fig. 5a). However, DIC was released into the overlying water at a much
higher rate (i.e. 3–4 times larger than $O_2$ consumption rate, Fig. 5b). The marine location (sediment
115) showed a larger DIC efflux than the riverine location (sediment 115), but the difference was
insignificant ($p > 0.05$, Student's $t$-test). Additionally, the $CH_4$ efflux was one to two orders of
magnitude smaller than the $O_2$ and DIC fluxes and showed significant differences between two
contrasting locations: the $CH_4$ efflux at the river location was more than five times higher compared to
the marine location (Fig. 5c).

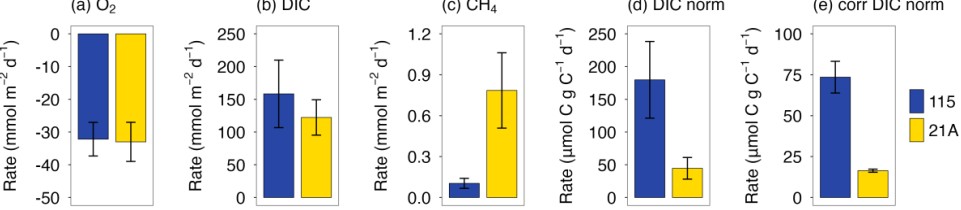


**Fig. 5.** Benthic fluxes of dissolved $O_2$ (a), DIC (b), and $CH_4$ (c) determined from whole-core incubation.
Positive and negative rates represent efflux (from sediment into overlying water) and influx (from overlying
water into sediment), respectively. Sediment TOC-normalized DIC (DIC norm) is presented in panel (d)
with TOC content being 2.2 wt.% for 115 and 5.0 wt.% for 21A. Panel (e) shows the OM-derived DIC,
corrected with DIN (see SI) and normalized by sediment TOC (corr DIC norm).

**3.5. Carbon emissions on bulk sediments**



During the aerobic incubation experiment, $CO_2$ accumulation was detected during the 3-hour rate
measurements for all timesteps. The $CO_2$ emission rate, expressed as µg C $g^{-1}$ $day^{-1}$, was the highest
at the start of the incubation. The rates dropped drastically in the first two weeks and then stabilized
after day 25. Here we present carbon emission rates at three timesteps representing the initial stage,
declining stage, and stable stage (Fig. 6). The silt-rich sediments showed both higher emission rates
throughout the incubation period (up to 120 µg C $g^{-1}$ $day^{-1}$) and stronger decreases in rate over time
(more than 60 µg C $g^{-1}$ $day^{-1}$), compared to sand-rich sediments (maximum rate around 35 µg C $g^{-1}$
$day^{-1}$; Fig. 6a). The TOC-normalized carbon emission rates were higher (up to three times) in the
three marine sediments (salinity 27–28) compared to the three riverine sediments (salinity 0–5)
throughout the experiment (Fig. 6b).

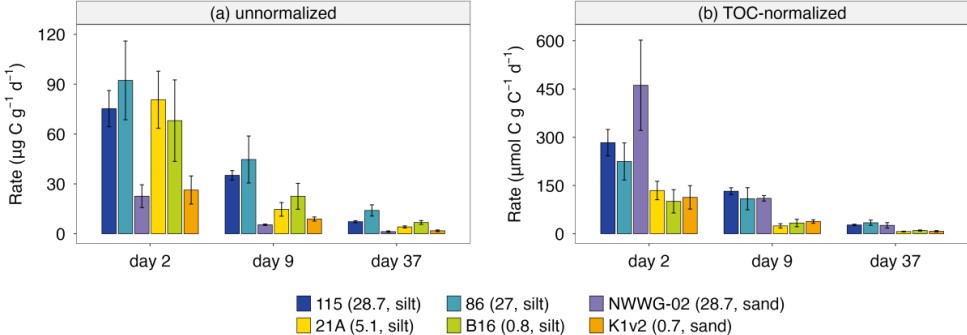


**Fig. 6.** Carbon emission rates in aerobic incubation at day 2, day 9 and day 37 from six sediments.
Note the different scales and units for the y-axis for unnormalized rate (a) and TOC-normalized rate
(b). Salinity and sediment texture are indicated in brackets in the legend.

The decreasing trend of $CO_2$ emission rate was also reflected in the cumulative percentage of
degraded TOC over time (Fig. 7), which increased fast initially and stabilized towards the end of the
incubation experiment. After the 37-day incubation period, the amount of degraded TOC ranged
between 1 to 7% for the investigated sites. Additionally, the percentage of degraded TOC was 2–4
times higher in sediments from marine locations than those in river locations, consistent with the
differences in carbon emission rates (Fig. 6b).

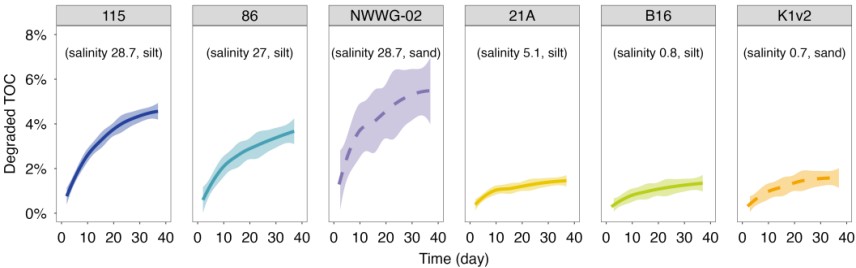




**Fig. 7.** The percentage of degraded TOC over time in aerobic incubation experiments. The shading
areas represent the 95% confidential interval for the fitted locally estimated scatterplot smoothing
(LOESS) curves.

**4. Discussion**
**4.1 Organic matter content, source and composition in estuarine sediments**
The PoR sediments are characterized by relatively high TOC contents compared to North Sea surface
sediments (0.03–2.79 wt.%; (Wiesner et al., 1990)), but in the range of Dutch coastal sediments (0–
9.8 wt.%; (Stronkhorst and Van Hattum, 2003)) or other harbor systems such as the Port of Hamburg
(2–7.6 wt.%; (Zander et al., 2020)). The high carbon contents arise from high productivity and rapid
burial of OM under high sedimentation rates; oxygen penetration is limited into rapidly accumulating,
organic-rich sediment and this most OM breakdown occurs via relatively slow, anaerobic processes
(Schulz and Zabel, 2006). Moreover, the fine sediment texture observed at most investigated sites will
limit oxygen diffusion and provides more sorption surface for OM (Keil et al., 1994), both contributing
to the preservation of sediment OM and thus high TOC content compared to sandy sediment. This is
expressed in the relatively low OM content of the coarser-grained sediments that were included in our
study (Fig. 2a). Besides the clear impact of grain size on OM content, we observed a general
decreasing trend in sediment TOC contents from river to marine area for PoR sediments, in line with
previous work on estuarine sediment OM (Strong et al., 2012).  The relatively low OM content in
sediment from the marine-dominated sites in part arises from the large input (up to 5.7 million tons per
year) in this area of repeatedly resuspended, OM-poor coastal sediment transported by strong tide
and waves (Cox et al., 2021). Furthermore, moving downstream from the riverine to the marine part of
estuarine systems, the contribution of OM-rich riverine sediment not only decreases but continuing
OM degradation in this transported sediment further diminishes riverine supply of OM from the
hinterland (Bianchi et al., 2018; Freitas et al., 2021). A confounding factor may be that OM burial and
degradation are not only affected by inputs and sediment properties as described above, but also by
the source and inherent properties of the OM.

The $\delta^{13}C_{org}$ and C/N ratio have been widely used to assess OM sources in coastal environments
(Canuel and Hardison, 2016; Lamb et al., 2006; Li et al., 2021; Middelburg and Nieuwenhuize, 1998).
The OM in the estuarine ecosystems can originate from multiple sources, and the typical ranges of
$\delta^{13}C_{org}$ and C/N ratio for the common OM sources are indicated in Fig. 2b. The trends in $\delta^{13}C_{org}$ and
C/N ratio suggest that OM in the PoR sediments is derived from a mixture of marine, riverine and
terrestrial OM that are sourced from algae, bacteria, soil OM, and terrestrial plants, the relative
contribution of these sources being a function of depositional conditions (riverine versus marine) as
reflected by salinity (Fig. 2b). The observed $\delta^{13}C_{org}$ values (−29–−23‰) and their trend against salinity
are similar to those in the broader Rhine estuary reported in earlier work (Middelburg and Herman,
2007), suggesting intense sediment reworking in connection with harbor expansion over the last 15
years have had little impact on sediment OM sources. Furthermore, the range in observed $\delta^{13}C_{org}$
values is lower than that reported for temperate marine OM (−18 and −22‰; (Thornton and



McManus, 1994)), reflecting a significant non-marine OM source even under nearly marine conditions
at the river mouth. Quantification of the different sources using end-member modelling similarly
indicates that the dominant OM source shifts with depositional environment: terrestrial OM in the most
river-dominated locations (up to 65%, salinity < 5), freshwater OM in the river-sea transitional area (~
45%, 5 < salinity < 25), and marine OM in the river-mouth area (up to 65%, salinity > 25).
Regarding the range of and trend in C/N values, it is important to note that the value is subject to OM-
specific alterations during sediment diagenesis: for higher plant litter, the C/N ratio decreases during
decomposition, while for aquatic detritus the C/N ratio increases during degradation (Hedges and
Oades, 1997; Wakeham and Canuel, 2006). These opposing diagenetic trajectories can result in a
convergence of C/N ratios of terrestrial and aquatic detritus (Middelburg and Herman, 2007). This
may explain bulk sediments at many of the investigated sites in the PoR research area have C/N
ratios near the upper limit of the typical range for freshwater algae (~8) or POC (~10), or around the
lower limit of the typical range for $C_3$ plants (~12, Fig. 2b). Compared to the C/N ratio, the BIT index is
thought to be less sensitive to diagenetic effects (Hopmans et al., 2004). This proxy indicates a
predominant riverine and/or terrestrial source of the sedimentary OM (Schouten et al., 2013). The BIT
values from this study are in line with the values previously determined by Herfort et al. (2006) in
sediment at Maassluis (0.74–0.82; close to NWWG-09, Fig 1), while they are much higher than those
determined in coastal sediments of the southern North Sea (0.02–0.25; (Herfort et al., 2006)),
highlighting the sharp transition in OM composition between estuarine and coastal systems and the
importance of non-marine OM throughout the harbor system.
The source proxies presented above ($\delta^{13}C_{org}$, C/N, BIT) indicate a strong terrestrial and riverine OM
signature across the salinity gradient in the PoR study area, with a considerable marine contribution
at the river mouth. The pyrolysis products from MOM can offer additional insights into sediment OM
sources and composition. Guaiacols and syringols are pyrolytic markers of terrestrial OM, as they are
characteristic structural moieties of lignin, a typical biopolymer of higher plants. Their relative
abundance together (7–28%) falls within the reported lignin fractions (3–57%) for various coastal
aquatic environments (Brandini et al., 2022; Burdige, 2007; Kaal et al., 2020). Although having
multiple potential sources, the markers of polysaccharides in our samples showed strong positive
correlations with both guaiacols ($R$ = 0.77, Pearson) and syringols ($R$ = 0.83, Pearson), suggesting
they were mainly derived from terrestrial higher plants. The decreasing trends of these markers
(relative abundance 10–40%) with increasing salinity, well aligned with $\delta^{13}C_{org}$ and BIT index, further
support the decreasing importance of terrestrial OM input towards the river month. In contrast, N-
compounds showed strong negative correlations with both guaiacols ($R$ = −0.84, Pearson) and
syringols ($R$ = −0.81, Pearson), suggesting a non-terrestrial OM origin such as protein from algal
detritus and chitin from various crustaceans (Nierop et al., 2017). *n*-Alkenes/alkanes, negatively
correlated with (terrestrial) polysaccharide-derived products ($R$ = −0.78, Pearson; Fig. S3), was
probably from non-terrestrial sources like algaenan (de Leeuw et al., 2006). The other detected
pyrolysis products constituted a major fraction (> 50%) but most correlated with all mentioned source



proxies moderately or poorly (−0.5 < $R$ < 0.5, Pearson; Fig. S3), thus are less effective as source
indicators as they likely originate from multiple, non-negligible sources.

All proxies and analytical techniques have their strengths and weaknesses in determining OM
sources. Here, we obtain further insight into MOM characteristics and the performance of various
techniques by exploring the relationships between different independent OM proxies and the end-
member modelling results. There is a striking agreement between the BIT index and the modelled
non-marine OM contribution ($R$ = 0.96, Pearson; Fig. 8a). The BIT index is a ratio that corresponds to
the relative importance of marine OM vs. soil and riverine OM. Its strong correlation with the modelled
non-marine OM (encompassing soil OM, riverine OM, and terrestrial vegetation input) suggests that
vegetation input was not a major component of the modelled non-marine OM contribution. Plant-
derived OM, however, was suggested to be a major MOM constituent, with an abundance of lignin-
derived products of up to 40% (Fig. 8b). Possibly, the lignin-derived products were mainly from eroded
soils carrying plenty of OM debris from the plants previously growing on them, or the amount of
vegetation input scaled proportionately with the amount of soil input.

The terrestrial OM fraction modelled from C/N and $\delta^{13}C_{org}$ showed a positive correlation with plant-
derived MOM pyrolysis products (Fig. 8c). Most data points seem to lie around the 1:1 curve except
two sand-rich outliers. However, interpreting their relationship in Fig. 8c is challenging because of the
complexity in assigning MOM pyrolysis products to terrestrial-derived OM in estuarine environment.
Phenols and N-compounds, partially derived from terrestrial OM, are not included in the presented
MOM-determined contribution here. On the other hand, pyrolysis of algal material also produces
polysaccharide-derived products (Stevenson and Abbott, 2019), which can lead to overestimation of
MOM-determined terrestrial contribution. Nevertheless, our study suggests using bulk proxies (C/N,
$\delta^{13}C_{org}$) in combination with biomarker proxies (BIT index, MOM pyrolysis products) can provide a
more complete picture of OM composition in highly dynamic systems like estuaries.

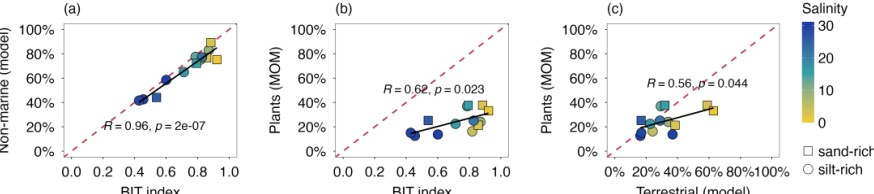

Fig. 8. Scatter plots of proxies for OM source: (a) BIT index vs. non-marine OM contribution (i.e.
terrestrial and riverine input from the three end-member modelling), (b) modelled terrestrial OM
contribution vs. plant-derived MOM pyrolysis products (i.e. sum of guaiacols, syringols,
polysaccharide-derived products), (c) BIT index vs. plant-derived MOM pyrolysis products (i.e. sum of
guaiacols, syringols, polysaccharide-derived products). The red dashed lines are 1:1 curves and the
black lines are the linear regression fitting curves.



**4.2 Organic matter degradation: rates and pathways**

In the 8-h whole-core incubation experiment, oxygen consumption was mostly due to OM mineralization; calculation of upward diffusive fluxes of reduced elements that can react with oxygen (e.g. $Fe^{2+}$, $Mn^{2+}$, $H_2S$) indicated that this represented a negligible oxygen sink at the sediment-water interface (< 1% of total oxygen uptake; see SI). The measured benthic $O_2$ consumption rates were very similar for sediments from two strongly contrasting environments in the marine and riverine part of the research area (Fig. 5a). The PoR sediments exhibited similar $O_2$ consumption rates ($33\pm6$ mmol $m^{-2}$ $d^{-1}$) as coastal North Sea sediments ($22.1\pm0.6$ mmol $m^{-2}$ $d^{-1}$; (Neumann et al., 2021)) and human-influenced estuarine sediment (27–82 mmol $m^{-2}$ $d^{-1}$; (Kraal et al., 2013)). Generally, sediment oxygen consumption decreases with increasing water depth from $45\pm22$ mmol $m^{-2}$ $d^{-1}$ on the inner shelf to $0.8\pm0.8$ mmol $m^{-2}$ $d^{-1}$ on the abyssal seafloor due to the increasing fraction of recalcitrant OM at the deeper realm (Jørgensen et al., 2022). However, when considering OM content in the surface sediment, the $O_2$ consumption rate for sediment 115 (depth ~ 25 m, TOC 2.2 wt.%) is about twice that of sediment 21A (depth ~13 m, TOC 5.0 wt.%). This suggests that OM source and composition as function of the depositional environment plays a key role in determining carbon oxidation rates and thereby the functioning of estuarine systems as important $CO_2$ sources in the global carbon cycle (Li et al., 2023).

Like $O_2$ consumption rates, DIC effluxes from the sediment were similar for the two contrasting sites (Fig. 5b). However, sediment 115 exhibited a larger TOC-normalized DIC flux (Fig. 5d), likely due to the greater supply and burial of fresh OM caused by a faster burial rate (10–15 cm $yr^{-1}$) in comparison to sediment 21A (<10 cm $yr^{-1}$; (Cox et al., 2021). Besides, sediment 21A at the riverine side was suggested to be richer in the eroded (ancient) soil OM (Fig. 8), often more recalcitrant than freshly produced OM. The respiratory quotient (RQ), determined as the ratio between DIC outflux and $O_2$ influx, was notably higher in our estuarine sediments (3.75–5) than the typical range observed in marine sediments (0.69–1.31; (Jørgensen et al., 2022), probably because carbonate dissolution enhances the DIC flux. Correction using DIN flux (Fig. S4) and Redfield ratio (C:N = 106:16) revealed that only about 40% of DIC was generated from OM degradation (see SI), among which 50–71% was produced aerobically. The RQ remained relatively high (1.4–2) after DIN correction, highlighting the importance of anaerobic degradation in shallow coastal systems, compared to the open ocean where RQs are often less than 1 (Jørgensen et al., 2022).

Regarding the role of estuaries in carbon cycling, a crucial transition in anaerobic OM degradation pathways is the onset of methanogenesis, which occurs when other TEAs have become depleted. Due to a lower salinity and thus sulfate concentration, sediment from a river location (21A; salinity 5.1) exhibited an eight-time larger $CH_4$ efflux (Fig. 5c) compared to the marine location (115; salinity 28.7) despite of less degradable OM with a stronger terrestrial signature (Fig. 2) as evidenced by the above-described lower OM mineralization rates relative to TOC content. Similar spatial variability of benthic $CH_4$ fluxes as function of salinity was documented in other estuaries but with rather different values (Gelesh et al., 2016; Li et al., 2021; Middelburg et al., 2002). The benthic fluxes measured



here do not directly translate into atmospheric $CO_2$ and $CH_4$ emissions as various processes (e.g.
carbonate system equilibria, $CH_4$ oxidation) act on the speciation and concentration of these
greenhouse gases released from the sediment. Nevertheless, estuaries are considered as hotspots
for both $CO_2$ and $CH_4$ emissions into atmosphere (Li et al., 2023; Middelburg et al., 2002). Therefore,
elucidating how in addition to OM content the source and composition as well environmental
conditions during OM degradation control the magnitude and speciation of carbon release from
estuarine sediment is important to better constrain the role of estuaries in global carbon cycling.

**4.3 The impact of perturbation on organic matter degradation**
Sediment dredging and its further management, such as relocation on land, often alter OM
degradation conditions substantially by reintroducing $O_2$. In principle, aerobic degradation is more
effective than anaerobic degradation as aerobic oxidation has a relatively high energy yield, especially
compared to sulfate reduction (Hansen and Blackburn, 1991). This is reflected in our whole-core
incubation results (Fig. 6) where aerobic mineralization (usually only a few millimeters thick;
(Revsbech et al., 1980)) accounted for 50–71% of the total OM-derived DIC production (~15 cm). By
manually perturbing sediments and exposing them to atmospheric oxygen in subaerial incubations,
we found that the initial (day 2) TOC-normalized carbon emission rate ($283\pm42$ μmol C g $C^{-1}$ $d^{-1}$for
115, $134\pm29$ μmol C g $C^{-1}$ $d^{-1}$ for 21A; Fig. 6b) increased to 3.8–8.4 times of that in undisturbed whole-
core incubation ($74\pm10$ μmol C g $C^{-1}$ $d^{-1}$for 115, $16\pm1$ μmol C g $C^{-1}$ $d^{-1}$ for 21A; Fig. 5e). These findings
agree with a slurry incubation experiment under contrasting redox conditions using Dutch coastal
sediments conducted by (Dauwe et al., 2001), which showed that the mineralization rate under
aerobic conditions was faster than anaerobic condition by up to one order of magnitude. Furthermore,
the increase in carbon emission rate was more pronounced in the riverine sediment (21A) with a
~740% increase after perturbation, compared to the marine sediment (115) with a ~280% increase.
We attribute this to the stronger terrestrial, recalcitrant signature of OM in the riverine part of the
investigated harbor area. (Hulthe et al., 1998) suggested that the impact of redox conditions and
specifically oxygen availability is greatest for relatively recalcitrant OM; fresh, labile OM is degraded
relatively rapidly under aerobic and anaerobic conditions. Therefore, the difference in the observed
rate increase following sediment perturbation may be attributed to the more active enzymatic catalysis
involved in the degradation of terrestrial OM, such as lignin, cellulose, and tannins (Hedges and
Oades, 1997), compared to freshly produced marine OM was more predominant. These OM source-
dependent differences in OM degradation rates were expressed across the six investigated sites: the
TOC-normalized carbon emission rates were over 100% higher in marine sediments (115, 86,
NWWG-02) than riverine sediments (21A, B16, K1v2) at almost all timesteps (Fig. 6b). This observed
difference is supported by our OM end-member analysis: sediments near the river mouth (115, 86,
NWWG-02) were composed of more than 50% marine OM and less than 20% terrestrial OM, whereas
sediments from the river side (21A, B16, K1v2) were dominated (>70%) by non-marine OM (Fig. 2c,
Table S2). The faster degradation rate of marine OM, such as algae, which was reported to be up to
10 times as quicker as terrestrial OM (Guillemette et al., 2013), likely explains the higher TOC-
normalized carbon emission rates in marine sediments.






623 In addition to the degradation rate, the extent of OM degradation is also affected by the OM source

624 and composition. By the end of the subaerial incubation experiment, marine sediments (115, 86,

625 NWWG-02) exhibited 2–4 times larger fractions of degraded TOC than riverine sediments (21A, B16,

626 K1v2; Fig. 7). Despite a lower TOC content, marine sediments contained a higher percentage of

627 fresher and more labile OM, thus resulting in a larger biodegradation fraction after 37 days of

628 subaerial incubation. Interestingly, sand-rich sediment NWWG-02 exhibited a notably larger

629 biodegradable OM fraction (up to 7%; Fig. 7), highlighting sediment texture may play an important role

630 besides OM sources. Silt-rich sediment can contain 20 times more mineral-associated OM  than

631 sand-rich wetland soils (Mirabito and Chambers, 2023). This mineral-associated OM, physically

632 protected by inorganic matrices from mineralization, was suggested to play a key role in lasting

633 carbon sequestration globally (Georgiou et al., 2022).

634

635 Despite variations in the fractions of degraded TOC, more than 90% of the organic carbon remained

636 in the sediments by the end of the 37-day aerobic incubation experiments (Fig. 7). This aligns with

637 other studies where a majority fraction (> 80%) of organic carbon remained preserved in sediments or

638 soils after prolonged incubation periods ranging from weeks to years (Gebert et al., 2019; Haynes,

639 2005; Plante et al., 2011). The predominant fraction of sediment OM being less degradable on such

640 timescales fits well with the relatively large amounts (~50%) of pyrolysis products derived from

641 (terrestrial) polysaccharide, *n*-alkenes/alkanes from algaenan, guaiacols and syringols from lignin.

642 However, (Zander et al., 2022) indicated that the slow degradation of the majority of OM could also be

643 attributed to its association with sedimentary minerals. Importantly, the remaining OM, while resistant

644 to degradation over weeks to years, is still potentially degradable on longer timescales and relevant

645 for the carbon footprint of perturbing estuarine sediment over decades. While our and other results

646 indicate that reintroduction of $O_2$ leads to a short-lived increase in estuarine OM degradation rates,

647 the degradation can still be stimulated under certain conditions. For instance, the addition of fresh,

648 readily degradable OM, known as priming, was reported to increase the degradability of old,

649 recalcitrant OM by 59% (Huo et al., 2017). This highlights the organic carbon turnover rate is rather

650 complex and can vary markedly under different sediment management practices.

651

652 **4.4 Implications and future perspectives**

653 Estuaries are sites of high OM production and processing, and understanding biogeochemical

654 processes within these regions is key to quantify organic carbon budgets along the river-estuary-

655 coastal ocean continuum (Canuel and Hardison, 2016). The use of multiple proxies (e.g. C/N, $\delta^{13}C_{org}$,

656 biomarkers) can improve our ability to understand, quantify, and predict the fate of organic carbon

657 delivered from continents to the oceans. Our study demonstrated that OM degradation exhibited a

658 source-specific pattern where both degradation rate and biodegradable pool varied over few times

659 depending on the origin of the OM. Degradation of OM is responsible for the recycling of essential

660 nutrients, for the oxygen balance of the aquatic system and its sediments and for most early

661 diagenetic processes (Middelburg et al., 1993). Recognizing and differentiating OM reactivity of



varying sources can help to refine the biogeochemical processes and minimize the uncertainty in
estimating OM mineralization and preservation efficiency in both field and theoretical frameworks.
Anthropogenic perturbation like dredging within the coastal zone have greatly intensified in recent
decades. Thus, it is crucial to recognize and quantify the impact of these sediment rework on carbon
mineralization. The sediment disturbance could have a considerable impact on the local carbon cycle
by accelerating the release of both $CO_2$ and $CH_4$ into the atmosphere (van de Velde et al., 2018).
Generally, the dredged material is relocated either underwater or on land. Exposure of sediment to
oxygenated environment can notably accelerate OM mineralization. When dredged sediment is
applied on land, the loss of the overlying water reduces the retention capacity of DIC, thereby
increasing $CO_2$ outgassing into atmosphere. Methane, a strong greenhouse gas, is often
oversaturated in the OM-rich coastal sediments where $CH_4$ bubbles are formed. Depending on the
dredging depth and sediment quality, dredging can lead to a short-term $CH_4$ emission peak by
increasing diffusion and ebullition (Maeck et al., 2013; Nijman et al., 2022). Estuarine systems are
characterized by a strong salinity gradient with a large variability of the depth of the sedimentary
methanic zone. Anaerobic oxidation of methane consumes approximately 71% of the $CH_4$ in marine
sediments (Gao et al., 2022), while dredging will inevitably disrupt anaerobic methane oxidation.
Further research should quantify the effect of dredging on $CH_4$ emission under realistic, large scale
dredging practices. Whether dredging activities can change the ecological service of estuarine
sediment from a carbon sink into a carbon source depends on the initial sediment carbon dynamics
as well as the intensity of human disturbance. Indubitably, estuaries will remain vulnerable to human
pressure and climate change. These alternations will in return influence the important drivers of the
estuarine, further affecting the balance between OM degradation and preservation (Heckbert et al.,
2012).

**Conclusions**
•   Applying multiple proxies from independent analyses gained a more comprehensive picture of the
OM sources and composition in highly dynamic environments such as estuaries. Here, the
combination of CN proxies, BIT index, and flash pyrolysis of MOM suggested the increasing
marine input and decreasing riverine and terrestrial input along a salinity gradient in the PoR area
located in a major European estuarine system. Throughout the salinity transition from freshwater
to nearly marine condition, considerable marine, riverine, and terrestrial (mostly eroded soil)
signals were detected at all locations. Consistency between the BIT index and CN based end-
member modelling suggested the robustness of the CN proxies and lipid biomarker in
distinguishing marine and non-marine OM contributions, whereas pyrolysis products of MOM are
suitable for assessing relative abundances of certain compounds (e.g. lignin) in a coastal
environment.
•   The PoR sediments, like many other coastal sediments, exhibited relatively high OM content and
reactivity, probably because of the high primary production and rapid sedimentation rate in these



shallow aquatic systems. However, TOC-normalized carbon release/production rates showed OM
degradation was significantly faster in marine sediments than in riverine sediments under both
intact conditions and disturbed oxygenated conditions, suggesting marine OM was more
susceptible to degradation. This is likely because marine sediment contained larger amount of
recently formed more labile OM. Higher sedimentation rates at the marine side contribute to a
better preservation of relatively fresher OM. In contrast, riverine sediments contain larger amounts
of eroded, ancient soil OM, possibly decades to centuries old, with only the more recalcitrant
fraction surviving by the time they reach the investigated locations. It highlights OM source and
composition, age, and sedimentation rate might be the key to the systematic explanation of OM
susceptibility in our study area.
• Nevertheless, OM degradability is not an inherent characteristic of OM itself, but also dependent
on the environmental context. Sediment perturbation with $O_2$ reintroduction showed to
substantially increase OM degradation by 2.8–7.4 times compared to intact conditions. With the
increasing intensity in coastal engineering, it is important to recognize the need to apply carbon-
sensitive management for sediments. Despite the degraded TOC being only 1–7% after 37-day
oxic incubation, the remaining organic carbon can still be turned over under certain conditions.
Therefore, quantifying organic carbon stability requires consideration of the relevant timescale as
well as that of the fast-changing environmental conditions.






**Author contribution**

GW conceptualized the study, developed the methodology, conducted the investigation and formal analysis, created visualizations, and wrote the original draft of the manuscript. KN and BY contributed to the investigation and formal analysis, and reviewed and edited the manuscript. SS and GJR reviewed and edited the manuscript. PK supervised the project, contributed to the conceptualization and methodology, acquired funding, and reviewed and edited the manuscript. All authors reviewed and agreed on the final version of the manuscript.

**Data availability**

The datasets used in this study are available from the corresponding author upon reasonable request.

**Declaration of competing interest**

The authors declare no competing interests.

**Acknowledgements**

This study is part of the project 'Transforming harbor sediment from waste into resource' funded by the Exact and Natural Sciences domain of the Dutch Research Council, NWO (grant number TWM.BL.019.005). We extend our gratitude to the Port of Rotterdam Authority, particularly Marco Wensveen and Ronald Rutgers, and Heijdra Milieu Service B.V. for their assistance with sediment collection. We thank Julia Gebert from Delft University of Technology for her stimulating discussion. We appreciate the scientific and technical staff from NIOZ Royal Netherlands Institute for Sea Research for their analytical support.

**Appendix A. Supplementary data**

The online version contains supplementary material available at XXX.



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
