# Peer review of "Title: Perturbation increases source-dependent organic matter degradation rates in estuarine"

_EGUsphere, 2024_

## Author Comment (AC1)

**We thank Reviewer 1 for taking time to give their thorough feedback and useful comments on our manuscript. The text in red indicates our response and the proposed modifications to the manuscript.**

**Reviewer 1:**

The manuscript focuses on organic matter degradation rates in the Amsterdam harbor estuary according to strong anthropogenic influence through dredging activities, using spatial monitoring data and diverse incubation processes. The research is very interesting and meaningful for carbon cycle and greenhouse gas emission from the sediment in such impacted area. The topic of this manuscript fits well with the journal's scope, and the data collected highlighted a strong sampling effort and figures are of good quality.

However, it is important to address some issues in the manuscript before acceptance for publishment, here are some specific comments:

We sincerely thank the reviewer for their careful and constructive feedback during the first review of our manuscript.

Line 80-82: more details in which way shifting salinity affected CH4 are needed (even if discussed in discussion, see comment "line 675-677").

We understand the importance to explain shifting salinity affects $CH_4$, but Reviewer 2 suggests to avoid highlighting $CH_4$ in this sentence for a smoother transition. Thus, we propose that we delete "*particularly for CH₄*" in this sentence to keep this transition more general, but we explain the salinity impact on $CH_4$ in the following sentence. The revised text is as:

> *This can lead to a strong spatial variability in OM degradation pathways and carbon dynamics, particularly for CH₄ (Cao et al., 2021). For example, SO₄²⁻ (usually higher in saline seawater) inhibits CH₄ production via multiple mechanisms (e.g. anaerobic oxidation of CH₄), which can lead to lower CH₄ emissions (Lovley et al., 1987; Egger et al., 2018).*

Line 87: reference is missing.
Line 606: no () for the reference. Dauwe et al. (2001)
Line 642: Zander et al. (2022)
Corrections will be made in the revised manuscript.

Line 89-92: yes, but give examples of naturally and anthropogenically induced sediment disturbance.
We will add examples in the revised manuscript.

Line 111: avoid terms like "our". Here use "the". To be corrected throughout the manuscript (e. g. "our study" replaced by "the present study"…).
Line 139: "our"?
We will change this throughout the revised version of the manuscript.

Line 122-123: out of context here…

This is also related to the next comment raised by Reviewer 1. We propose to make the following modifications to this paragraph in the revised manuscript:

*Understanding the processing of OM within estuaries takes on further importance because many estuarine systems are intensively altered by human activities (Arndt et al., 2013; Heckbert et al., 2012; Holligan and Reiners, 1992). To increase or maintain waterway navigability, dredging is  commonly practiced in many coastal regions and rivers worldwide. More than 600 million m³ of dredged material is generated annually just in Western Europe, China, and the USA (Amar et al., 2021).  While the dredged sediments are often treated as waste and disposed at sea, there is a growing trend of reusing dredged sediments on land, such as beach nourishment, habitat restoration, and land reclamation (Brils et al., 2014), aiming to valorize these massive amounts of dredged materials. However, perturbations during dredging and the following management practices pose a great unknown in the fate of the large amount of organic carbon stored in these sediments, with oxygen exposure potentially leading to a substantial increase of carbon release (LaRowe et al., 2020). Given the needs of sediment dredging and the sustainable management of these materials (van de Velde et al., 2018), it is of great importance to understand to what extent anthropogenic sediment perturbations may potentially affect OM processing and carbon emissions from estuarine sediments.*

Line 138-139: representative of dredged sediment conditioned on land? Is this conditioning the major process for dredged sediment? Line 151 mentioned sediment relocated in the sea. More explanations are needed to justify the choice of these open-air incubations.

We appreciate the reviewer's comments regarding the rationale of the open-air incubations. We will add detailed explanations to justify the choice of open-air incubations in the revised manuscript.

Indeed, as mentioned in Line151, most of the dredged sediments are currently relocated to the North Sea, and the whole-core incubations were performed under the water. However, long-term incubation experiments with submerged sediments (including slurries) may suffer from kinetic limitations in oxygen supply, potentially introducing artefacts in estimating degradation rates. Open-air incubations with optimally wetted sediments, however, avoid these kinetic limitations, allowing us to better quantify the oxic degradation potential.

Another reason is linked to the previous comment of Reviewer 1 (see above). There is a growing trend and need to valorize dredged sediment on land (e.g. land reclamation, habitat restoration, beach nourishment) rather than directly disposing of these materials at sea. We recognize that the choice of open-air incubation in this study is a simplification of those on-land applications and may not fully represent the reality, but the findings of open-air experiments are still insightful in estimating the carbon emission potential of dredged sediments when exposed to oxygen on land or in oxygenated aquatic environments.

Line 151-152: The rates shown are per year?

Yes.

Line 155-156: how many cores replicates for each location?

Bulk sediments were collected using one sediment core (Line 155–156), while triplicate cores were used for the intact incubation (Line 241). Clarification will be added.

Line 242-243: Even evident for the author, precise the reason to have 20 cm of overlying water.

The reason to keep this amount of overlaying water is to ensure adequate solute concentration changes during incubation while minimizing the sampling impacts of the overlaying water sampling. Each time, we sampled 30-mL overlaying water while the same volume of site water was automatically refilled from a reservoir. In this way, we keep a relatively small fraction (i.e. 2.3%) of water replacement at each sampling time. We will add an explanation in the revised manuscript.

Line 285: Why 37 days incubation period were chosen?

Thanks for pointing this out. This length of incubation time was decided during the progress of the incubation experiments, based on the observed carbon dynamics. The carbon emission rates on day 23, 30 and 37 were much smaller and more stable compared to the rates measured at the beginning (Fig. 6). These data points allow us to extrapolate the percentage of degradable TOC from day 37 onwards (Fig. 7), and the carbon release from day 37 onwards is likely associated with the slow-carbon pool, which is relatively stable. A prolonged incubation is unlikely to add substantial significance to the data interpretation. An explanation will be added.

Line 581: Short range of these different values is needed.

This will be added.

Line 675-677: include the concept of SMTZ (sulfate-methane transition zone)

We will include it in the revised manuscript.

Line 681: "depending on"

We will correct it.

Line 687: I don't know if the conclusion needs to be so precise about the results of the study…

Thank you for pointing this out. We will revise the conclusion by removing unnecessary details.

Additional comments: where are the dredging locations on Fig. 1? Dredging affects the upstream and downstream areas of the estuary in a similar way?

Dredging is performed throughout the port area in Fig. 1, but there are more dredging activities at the marine side than at the riverine side. Information will be added to section 2.1.

Please give some details on this subject. What would be the consequences of such an imbalance, according to quantity of sediment dredged in each zone?

We appreciate the reviewer mentioning this point. Indeed, we did not explore the impact of this aspect in the manuscript. We will include this discussion in the revised manuscript. More dredging activities at the marine site (115) might contribute to its lower TOC content (Fig. 2a). However, higher fractions of labile OM (Fig. 7) and faster degradation rates (Fig. 5, 6) in marine sediments suggest that the OM quality/degradability in the investigated area is mainly controlled by the OM input sources and the rapid sedimentation, while dredging, although enhancing degradation of labile OM, is not the determining factor of OM quality/degradability.

What about the sediment relocated in the sea in terms of potential carbon mineralization processes?

We appreciate this consideration. In the Port of Rotterdam (PoR), the current practice of relocating dredged sediment to the North Sea may enhance the OM mineralization rate (3.4–7.6 times) similarly like open-air experiments due to exposure to oxygenated seawater. However, the degree of mineralization is affected by the sediment dynamics after disposal (i.e. sediment resuspension might result in more extensive mineralization, while rapid sediment burial, which limits oxygen exposure, can better preserve organic carbon). We will discuss this in the revised manuscript.

And finally CO2/CH4 effluxes toward the atmosphere?

Thank you for the inspiration, but we feel this is beyond the scope of this manuscript. This study pivots on perturbation impacts on OM mineralization coupled to OM composition. While the eventual outgassing is not the focus here, the measured carbon release from undisturbed and disturbed sediments can potentially enter the atmosphere. We will briefly mention this in the revised manuscript as a possible direction for future research.

**References**

Aller, R. C., Blair, N. E., Xia, Q., and Rude, P. D.: Remineralization rates, recycling, and storage of carbon in Amazon shelf sediments, Cont Shelf Res, 16, 753–786, https://doi.org/10.1016/0278-4343(95)00046-1, 1996.

Amar, M., Benzerzour, M., Kleib, J., and Abriak, N. E.: From dredged sediment to supplementary cementitious material: characterization, treatment, and reuse, International Journal of Sediment Research, 36, 92–109, https://doi.org/10.1016/j.ijsrc.2020.06.002, 2021.

Arndt, S., Jørgensen, B. B., LaRowe, D. E., Middelburg, J. J., Pancost, R. D., and Regnier, P.: Quantifying the degradation of organic matter in marine sediments: A review and synthesis, Earth Sci Rev, 123, 53–86, https://doi.org/10.1016/j.earscirev.2013.02.008, 2013.

Brils, J., de Boer, P., Mulder, J., and de Boer, E.: Reuse of dredged material as a way to tackle societal challenges, J Soils Sediments, 14, 1638–1641, https://doi.org/10.1007/s11368-014-0918-0, 2014.

Cao, C., Cai, F., Qi, H., Zhao, S., and Wu, C.: Differences in the sulfate–methane transitional zone in coastal pockmarks in various sedimentary environments, Water (Switzerland), 13, 1–18, https://doi.org/10.3390/w13010068, 2021.

Heckbert, S., Costanza, R., Poloczanska, E. S., and Richardson, A. J.: Climate Regulation as a Service from Estuarine and Coastal Ecosystems, Elsevier Inc., 199–216 pp., https://doi.org/10.1016/B978-0-12-374711-2.01211-0, 2012.

Holligan, P. M. and Reiners, W. A.: Predicting the Responses of the Coastal Zone to Global Change, Adv Ecol Res, 22, 211–255, https://doi.org/10.1016/S0065-2504(08)60137-3, 1992.

LaRowe, D. E., Arndt, S., Bradley, J. A., Estes, E. R., Hoarfrost, A., Lang, S. Q., Lloyd, K. G., Mahmoudi, N., Orsi, W. D., Shah Walter, S. R., Steen, A. D., and Zhao, R.: The fate of organic carbon in marine sediments - New insights from recent data and analysis, Earth Sci Rev, 204, 103146, https://doi.org/10.1016/j.earscirev.2020.103146, 2020.

van de Velde, S., van Lancker, V., Hidalgo-Martinez, S., Berelson, W. M., and Meysman, F. J. R.: Anthropogenic disturbance keeps the coastal seafloor biogeochemistry in a transient state, Sci Rep, 8, 1–10, https://doi.org/10.1038/s41598-018-23925-y, 2018.

---

## Author Comment (AC2)

**We thank Reviewer 2 for the detailed and constructive feedback. The text in red indicates our response and the proposed modifications to the manuscript.**

**Reviewer 2:**

"Perturbation increases source-dependent organic matter degradation rates in estuarine sediments" by Guangnan Wu et al. is a comprehensive and well-executed study investigating the chemical characteristics of sediment organic matter in Rotterdam harbour and the potential response of sediment carbon stocks to human disturbances (i.e. dredging). The study employs a range of techniques in the basic characterization of samples along a salinity transect (bulk sediment C/N, del13C, BIT index and pyrolysis-GCMS analysis of macromolecular organic matter) followed by a set of incubation experiments (both classical whole-core and homogenized sediment subaerial set-ups) to show how remineralization processes respond to disturbance. The data is of high quality and is generally processed and interpreted well, with a high degree of integration between the various lines of evidence. Overall, the outcome is convincing and should be published. However, I outline below one key issue related to the data processing that requires more careful consideration in the text, and several minor comments that should be addressed.

We sincerely thank the reviewer for their positive recommendation as well as the constructive feedback on our manuscript.

Key issue:

In processing the results of the whole-core incubations, ammonium fluxes are used to estimate the fraction of the DIC flux that is derived from organic matter remineralization (results shown in Fig. 5e). This calculation assumes Redfield stochiometry for the degrading material, i.e. ammonium and DIC from remineralization are released in the ratio 106:16. This may be valid for the marine end-member site 115 but not necessarily for riverine end-member site 21A. Considering the bulk sediment C/N ratio data it is likely that site 21A releases more DIC per mole ammonium, even if the most reactive fraction of the sedimentary organic matter is relatively nitrogen rich. I suggest that this uncertainty is somehow included in the calculations, for example by presenting additional bars in Fig. 5e or a summary table. This issue is important because the quantification of remineralization-derived DIC production in the whole-core incubations is used later in the study when comparing with rates in the subaerial incubations, ie. assessing by how much dredging of sediment onto land would stimulate remineralization.

We appreciate the reviewer for pointing this out. We propose to make changes to Fig. 5e (see below). Indeed, the organic matter (OM) composition and C/N ratio differ between site 115 and site 21A (Table 2 below). Therefore, a different C/N ratio should be used when converting ammonium flux to OM-mineralized DIC flux (Fig. 5e). We will use different C/N ratios (with literature-reported ranges, Table 2) to account for OM compositional variability between site 115 and 21A and their corresponding uncertainties will be presented in a summary table (as Table 2). Besides the modification of Fig. 5e, we will carefully check and change the quantifications associated with Fig. 5e throughout the manuscript.

Proposed new Fig. 5, supplemented by Table 2:

[Figure]

**Fig. 5.** Benthic fluxes of dissolved $O_2$ (a), DIC (b), and $CH_4$ (c) determined from whole-core incubation. Positive and negative rates represent efflux (from sediment into overlying water) and influx (from overlying water into sediment), respectively. Sediment TOC-normalized DIC (DIC norm) is presented in panel (d) with TOC content being 2.2 wt.% for 115 and 5.0 wt.% for 21A. *Panel (e) shows the DIN-corrected DIC fluxes (representing OM-derived DIC), normalized by sediment TOC (corr DIC norm), using minimum (min) and maximum (max) C:N ratios from literature (see Table 2). Error bars represent the standard deviation from triplicate core incubations.*

*Table 2. The bulk C/N ratios at sites 115 and 21A compared to values reported in literature in marine and riverine sediments. The literature-reported ratios were used to calculate DIN-corrected DIC fluxes (Fig. S4c) and TOC-normalized, DIN-corrected DIC fluxes (Fig. 5e) for sites 115 and 21A.*

| Sediment | Site 115 (marine) | Site 21A (riverine) |
|---|---|---|
| Measured bulk C:N ratio | 7.2 | 9.9 |
| Literature-reported bulk C:N ratio in marine/riverine sediments | 6.625 (Redfield, 1958), 7.4 (Martiny et al., 2014), 5.6–7.5 (Körtzinger et al., 2001) | 12 (Benoit et al., 2006), 13.1 (They et al., 2017), 7.1–13 (Maranger et al., 2018), 10.4–14.2 (Hecky et al., 1993) |
| Measured DIN flux | $9.8 \pm 1.3$ mmol m$^{-2}$ d$^{-1}$ | $7.0 \pm 0.9$ mmol m$^{-2}$ d$^{-1}$ |
| Min C:N ratio for conversion | 5.6 | 7.1 |
| Max C:N ratio for conversion | 7.5 | 14.2 |
| DIN-corrected DIC flux (Fig. S4c), min mean (± SD)–max mean (± SD) | 54.7 (±7.2)–73.3 (±9.7) mmol m$^{-2}$ d$^{-1}$ | 49.8 (±6.5)–99.7 (±12.9) mmol m$^{-2}$ d$^{-1}$ |
| DIN-corrected DIC flux with TOC normalization (Fig. 5e), min mean (± SD)–max mean (± SD) | 62.2 (±8.2)–83.2 (±11.0) µg C g$^{-1}$ day$^{-1}$ | 17.6 (±1.0)–35.1 (±2.0) µg C g$^{-1}$ day$^{-1}$ |

Minor comments:

Line 81-82: No need to highlight CH4 specifically. The point is valid for the balance between any given set of remineralization pathways.

We agree there is no need to highlight $CH_4$ specifically in this sentence, but it is important to mention somewhere that shifting salinity affects $CH_4$ dynamics (suggested by Reviewer 1). So, we propose that we delete "particularly for $CH_4$" in this sentence to keep this transition more general and smoother (Line 81–82) and introduce salinity-$CH_4$ dynamics in the following sentence. The revision is as:

*This can lead to a strong spatial variability in OM degradation pathways and carbon dynamics (Cao et al., 2021). For example, $SO_4^{2-}$ (usually higher in saline seawater) inhibits $CH_4$ production via multiple mechanisms (e.g. anaerobic oxidation of $CH_4$), which can lead to lower $CH_4$ emissions (Lovley et al., 1987; Egger et al., 2018).*

Line 84-87: Consider rewording the sentence ("estuaries" appears to be used in both a general and a specific way in the same sentence) and add the reference.

Will be changed.

Line 88: Should be "penetration depth".

Line 99: Make sure to use brackets correctly.

Line 123: Should be "dredged sediment".

Line 137-139: Should be "bottle incubations"

Line 139: Should be "Our results show..." or "We show..."

Line 141: Should be "the properties of OM influence..."

Line 213: Remove "an"

Line 383: Should be "overlying water".

Line 438: Should be "thus" not "this"

Line 478: Should be "may explain the observation that".

Line 607: Check the formatting of the citation.

Line 620: Should be "10 times faster than..."

We will double check and correct the typos and grammar throughout the manuscript.

Line 162: Use "end-members" or similar, rather than "realm"

Will be changed.

Line 220-229: It would be useful for a reader trying to reproduce this method to know how the HF was disposed of during the protocol. Is it evaporated at some stage? This is a dangerous chemical and lab protocols need to be carefully designed.

The HF waste was collected in a high-density polyethylene plastic container designated for HF waste, further handled by the EHS team (i.e. ARBO staff in the Netherlands). Details will be added in the revised manuscript.

Line 229: Clarify that "MOM" here refers to the residue of the previous steps.

Clarification will be made.

Line 283: It is not clear what is meant by "60% water-filled volume". It is difficult to estimate the porosity of freeze-dried and homogenized sediment after re-wetting due to changes in grain size distribution during these processing steps.

We thank the reviewer for the insightful comment. Here, the "60% water-filled pore space" refers to 60% of the porosity of the freeze-dried, homogenized sediment. We agree that the grain size distribution may change after rewetting, which could further affect the sediment porosity. To avoid confusion, we propose to simply describe the materials as "rewetted sediments". For clarity, we will specify that the added volume of artificial rainwater corresponds to 60% of the porosity of the freeze-dried sediment.

Line 345: "Asterisk/stars" not "asteroids". Also unclear what the arrows in the plot indicate, should they be pointed in the opposite direction to link the numerical labels to the 2D fields in the x-y plot?

Will be changed.

Line 245: Should be Table S3, not S2.
We will double check the ordering of tables and figures.

Fig. 6: Please add the information about "marine" vs. "riverine" stations to the plot or legend.
Will be added.

Fig. 7: What do the stipple vs. solid lines indicate? Also show which sites are "marine" and "riverine".
Different line styles were intended to differentiate two sediment textures (i.e. sand-rich sediment vs. silt-rich sediment). To avoid confusion, we will use the solid lines for all subplots in Fig. 7. We will also specify in the new plot that site 115 is the "marine" site, and site 21A is the "riverine" site.

Line 451: This is a valid consideration rather than a confounding factor.
Will be changed.

Line 517: It is not unusual that soil OM contains similar chemical signatures to the higher plants that grow and decay in them. It is perhaps misleading to state that the correlation between BIT and the EMMM results implies low contribution of vegetation input, rather it should be stated that, as you conclude later, the plant signals are transferred via the soils (where they also pick up the BIT signal).
Thank you for noticing. The text will be revised as follows:

> *Its strong correlation with the modelled non-marine OM (encompassing soil OM, riverine OM, and terrestrial vegetation input) suggests that vegetation input was not a major component of the modelled non-marine OM contribution. Plant-derived OM, however, was suggested to be a major MOM constituent, with an abundance of lignin-derived products of up to 40% (Fig. 8b). Possibly, the lignin-derived products were mainly from eroded soils carrying plenty of OM debris from the plants previously growing on them, or the amount of vegetation input scaled proportionately with the amount of soil input., or if not, vegetation OM was transferred from the soils, where they pick up the (enhanced) BIT signal. The latter assumption was supported by the MOM constituents, where the relative abundance of lignin-derived products (indicating higher plant signals) was up to 40% (Fig. 8b).*

Line 551-552: The comparison with ocean margin trends is not useful here as the reactivity gradients in near-shore areas is more likely to be controlled by terrestrial inputs than by water-depth fractionation of sinking OM.
We agree with the reviewer. This comparison will be removed in the revised manuscript.

Line 564: Remove "ancient". There is no information about age of the material so this is not relevant.
Will be removed.

Line 567-568: Interpretation of carbonate dissolution needs to be expanded. Is there evidence for this in the data or literature?

Yes, the DIC flux/DIN flux was ~16, larger than the C/N ratio (5.6–14.2; Table 2) of marine and riverine OM. This suggests that other pathways, besides OM mineralization, contributed to the carbon release. In this case, carbonate dissolution was a likely process. We propose the following changes in the revised text.

> *The respiratory quotient (RQ), determined as the ratio between DIC outflux and $O_2$ influx, was notably higher in the investigated estuarine sediments (3.75–5) than the typical range observed in marine sediments (0.69– 1.31; Jørgensen et al., 2022), probably because carbonate dissolution enhances the DIC flux suggested by the DIC/DIN ratios (~16; Fig S4b), which was larger than the literature-reported C/N ratio of marine and riverine OM (5.6–14.2; Table 2), particularly for site 115 (marine).*

Line 577: The higher CH4 flux in a whole core incubation at a more freshwater site can be due to the higher position in the sediment column of the SMTZ. See e.g. the diagram in Fig. 7 of https://esd.copernicus.org/articles/13/633/2022/esd-13-633-2022.html.
Modifications will be made.

Line 633: Also refer to Keil et al. (1994).
Will be added.

Line 677-678: Separate the statements about global AOM estimates and dredging impacts, these are not obviously connected.
We propose the following revisions to the text in this paragraph:

> *Methane, a strong greenhouse gas, is often oversaturated in the OM-rich coastal sediments where $CH_4$ bubbles are formed. Most $CH_4$ is trapped below the sulfate-methane transition (SMTZ), and approximately 71% of the $CH_4$ is removed via anaerobic oxidation of methane (AOM) in marine sediments (Gao et al., 2022), but the removal efficiency is usually lower in estuarine sediments due to lower $SO_4^{2-}$ concentrations. Dredging, which can remove the SMTZ completely or partially, is likely to impact the functioning of the AOM filter, consequently leading to a short-term $CH_4$ emission peak by increasing diffusion and ebullition (Maeck et al., 2013; Nijman et al., 2022).  While dredging intensity and the SMTZ position can differ greatly within estuaries, further research should quantify the effect of dredging on $CH_4$ emission under realistic,  and site-specific dredging practices.*

**References**

Benoit, P., Gratton, Y., and Mucci, A.: Modeling of dissolved oxygen levels in the bottom waters of the Lower St. Lawrence Estuary: Coupling of benthic and pelagic processes, Mar Chem, 102, 13–32, https://doi.org/10.1016/j.marchem.2005.09.015, 2006.

Cao, C., Cai, F., Qi, H., Zhao, S., and Wu, C.: Differences in the sulfate–methane transitional zone in coastal pockmarks in various sedimentary environments, Water (Switzerland), 13, 1–18, https://doi.org/10.3390/w13010068, 2021.

Gao, Y., Wang, Y., Lee, H. S., and Jin, P.: Significance of anaerobic oxidation of methane (AOM) in mitigating methane emission from major natural and anthropogenic sources: a review of AOM rates in recent publications, Environmental Science: Advances, 1, 401–425, https://doi.org/10.1039/d2va00091a, 2022.

Hecky, R. E., Campbell, P., and Hendzel, L. L.: The stoichiometry of carbon, nitrogen, and phosphorus in particulate matter of lakes and oceans, Limnol Oceanogr, 38, 709–724, https://doi.org/10.4319/lo.1993.38.4.0709, 1993.

Jørgensen, B. B., Wenzhöfer, F., Egger, M., and Glud, R. N.: Sediment oxygen consumption: Role in the global marine carbon cycle, Earth Sci Rev, 228, https://doi.org/10.1016/j.earscirev.2022.103987, 2022.

Körtzinger, A., Hedges, J. I., and Quay, P. D.: Redfield ratios revisited: Removing the biasing effect of anthropogenic $CO_2$, Limnol Oceanogr, 46, 964–970, https://doi.org/10.4319/lo.2001.46.4.0964, 2001.

Maeck, A., Delsontro, T., McGinnis, D. F., Fischer, H., Flury, S., Schmidt, M., Fietzek, P., and Lorke, A.: Sediment trapping by dams creates methane emission hot spots, United States of America, Environmental Science and Technology, 8130–8137, 2013.

Maranger, R., Jones, S. E., and Cotner, J. B.: Stoichiometry of carbon, nitrogen, and phosphorus through the freshwater pipe, Limnol Oceanogr Lett, 3, 89–101, https://doi.org/10.1002/lol2.10080, 2018.

Martiny, A. C., Vrugt, J. A., and Lomas, M. W.: Concentrations and ratios of particulate organic carbon, nitrogen, and phosphorus in the global ocean, Sci Data, 1, https://doi.org/10.1038/sdata.2014.48, 2014.

Nijman, T. P. A., Lemmens, M., Lurling, M., Kosten, S., Welte, C., and Veraart, A. J.: Phosphorus control and dredging decrease methane emissions from shallow lakes, Science of the Total Environment, 847, 157584, https://doi.org/10.1016/j.scitotenv.2022.157584, 2022.

Redfield, A. C.: THE BIOLOGICAL CONTROL OF CHEMICAL FACTORS IN THE ENVIRONMENT, 205–221 pp., 1958.

They, N. H., Amado, A. M., and Cotner, J. B.: Redfield ratios in inland waters: higher biological control of C:N:P ratios in tropical semi-arid high water residence time lakes, Front Microbiol, 8, https://doi.org/10.3389/fmicb.2017.01505, 2017.

---

## Author Comment (AC3)

**We thank Reviewer 3 for taking time to give their feedback. The text in red indicates our response and the proposed modifications to the manuscript.**

**Reviewer 3:**

The manuscript of G. Wu et al. investigates the increase of organic matter (OM) degradation produced by anthropogenic perturbations such as dredging activity. The method stands on the elemental (C/N) and isotopic ($\delta$13C) characterization of organic matter of 49 sites from the Port of Rotterdam. Molecular characterization was done on 13 of the 49 sites, reoxydation experiments on 6 sites and in situ OM remineralization have been estimated on 2 sites. The authors conclude from their studies that Cl#1 - CN proxies are robust to identify contribution of marine versus non-marine OM sources, Cl#2 - marine OM is more labile than terrestrial OM and Cl#3-aerobic conditions accelerates OM remineralization compared to natural (mainly anaerobic) conditions.

From my point of view, this manuscript is not reaching the quality requirement to get published in BG : the rare new knowledges brought by this manuscript are only site specific. In the conclusion, the authors underline more general concepts (Cl#1, Cl#2, Cl#3), however it corresponds to already well-established concept in biogeochemistry that does not present significant novelty.

We sincerely thank the reviewer for their time and effort in evaluating our manuscript. However, we disagree with the assessment that the study lacks sufficient novelty or relevance for publication in *Biogeosciences*. This study presents a detailed characterization of estuarine OM coupled to degradation rates in-situ and perturbed, which is in our view novel. While some of the broader concepts discussed (e.g. selective preservation, aerobic mineralization) are indeed established in biogeochemistry, the contribution of this study lies in applying and contextualizing these concepts within a heavily disturbed/dredged estuarine system, which is insufficiently explored in the literature. Besides, the findings provide crucial insights into how human-induced disturbances interact with natural OM processing, with implications for carbon cycle and release. As estuarine systems facing increasing pressure of human perturbation, we believe this study represents a meaningful contribution to the field, as also noted by Reviewer 1 and 2.

In addition, a detailed reading of the manuscript reveal that the scientific hypothesis defended along the manuscript (that perturbation increases source-dependent organic matter degradation rates in estuarine sediment) is not properly argued. While the source characterization of the OM received large analytical efforts, and a valid interpretation supported by some modelisation, four others critical step of the demonstration seems not strong enough. First, and most important, the demonstration of a source and composition dependence of OM degradation rate is not convincing. The authors build their hypothesis from the literature and a reasonable intuition but that is in complete opposition with the dataset presented. Indeed, the two sites, selected because they present very contrasted OM origin and composition, present very similar O2 fluxes that is interpreted -reasonably- as a similar intensity of remineralization rate. This result should better be used to underline a limit of the observations from the literature. Instead, the authors use an artefact (normalization by the TOC) to build a contrast between

the two stations while the relation between quantity of OM and remineralization rate have not been presented. It seems more rational to conclude from their dataset that OM quantity does not play an important role in deposit, probably because it is in large excess.

We agree that the total amount of sediment organic carbon might be in large excess relative to $O_2$ availability, but we want to clarify that the $O_2$ fluxes represent the oxygen consumption of fresh OM deposited at the sediment surface, rather than the whole sediment cores, as the oxygen penetration depth is usually a few millimeters and even less in OM-rich coastal sediments (Cai and Sayles, 1996). We will add this point to the revised manuscript.

We do not agree with the suggestion that the use of the TOC-normalized rate is an artefact. Normalization on TOC is a commonly used approach to assess the relative reactivity of OM (e.g. Freitas et al., 2025; Zander et al., 2020, 2022), especially when TOC content differs considerably between sites (in this case, 2.2 wt.% and 5 wt.%). The TOC-normalization allows us to compare OM degradation efficiency between sites with different OM compositions (more fresh algal material at site 115 while more terrestrial-derived OM at site 21A) and, furthermore, between different incubation conditions (Fig. 5e and 6b). Importantly, it highlights that caution is required when interpreting bulk OM content in terms of $CO_2$ release potential.

Second in importance, there is no consideration on primary production, while freshly produced algea are probably more labile than any other carbon source. In other word the local recycling of carbon between photosynthesis and respiration is neglected in this manuscript while it is probably very important in this very shallow environment.

We would like to clarify that primary production and the associated input of labile algal OM are not neglected in the manuscript. We first mentioned it in the whole-core incubation experiments, where the local OM production and transport dominate (Line 561–565). Then we expanded this aspect to the discussion of carbon release from mixed, dredged sediments (Line 619–621). We later mentioned it again in the conclusion section (Line 704–710).

Third, while the introduction suggests a study covering many sites and embracing some spatial variability, the comparison between perturbated and not perturbated sediment is done only on 2 sediment, which seems not enough to support conclusions that could be generalized to others estuaries, and thus limits the interest of the results.

The study does show a comprehensive dataset on sediment OM properties throughout the harbor: 13 sites comprising OM characterization, 6 sites dealing with reoxidation. For further context, intact core incubations were performed for two contrasting sites. Altogether, this is a very elaborate dataset that links detailed sediment (OM) properties to rates of processes, which is novel and provides new insights into the $CO_2$ release potential of disturbed estuarine sediments. Furthermore, this study is conducted in one of the largest estuaries and the busiest harbor of Europe, which is under the impact of substantial anthropogenic disturbance. Given the growing trend of human modifications of coastal systems, we

believe the findings are relevant and will contribute to the further understanding of human impacts on the sequestration of coastal blue carbon.

Fourth, the methodology of perturbation does not seem adapted to an estuarine environment, since it corresponds to a not water-saturated sediment more adapted to mimic soil aeration than resuspension induced by dredging activities.

This was also noticed by Reviewer 1. We will add a detailed explanation in both the Introduction and Discussion sections of the revised manuscript. In short, the open-air incubation was chosen because: (i) it provides insights into potential rates of OM mineralization of sediment exposure to oxygenated water, while limiting kinetic constraints of $O_2$ transportation in slurry experiments, and (ii) it tests the impact of on-land application (which is an increasing trend in sediment management practices) on carbon emission.

Additionally, the dry freezing of the sediment before incubation would certainly modify the properties of the reactive organic matter – which have not been tested.

We acknowledge that freeze-drying may change the properties of the reactive OM. However, alternative methods like air-drying can also change OM characteristics, including the loss of the most labile carbon fraction during air-drying (which usually lasts a few days). Some incubation studies on soils/sediments were based on freeze-dried samples and reported limited impact of freeze-drying on carbon emissions (He et al., 2022; Wu et al., 2020). Fromin (2025) suggests that there is rarely consensus on a best practice when studying microbial processes in soils and sediments, and the appropriate methods often depend on the specific goals of the study. Given the limitations of different approaches and the aim to quantify the labile OM fraction, freeze-drying was considered the more suitable option for sample pre-treatment in our case.

**References**

Cai, W.-J. and Sayles, F. L.: Oxygen penetration depths and fluxes in marine sediments, Mar Chem, 52, 123–131, https://doi.org/https://doi.org/10.1016/0304-4203(95)00081-X, 1996.

Freitas, N. L., Walter Anthony, K., Lenz, J., Porras, R. C., and Torn, M. S.: Substantial and overlooked greenhouse gas emissions from deep Arctic lake sediment, Nat Geosci, https://doi.org/10.1038/s41561-024-01614-y, 2025.

Fromin, N.: Impacts of soil storage on microbial parameters, SOIL, 11, 247–265, https://doi.org/10.5194/soil-11-247-2025, 2025.

He, Y., Zhang, T., Zhao, Q., Gao, X., He, T., and Yang, S.: Response of GHG emissions to interactions of temperature and drying in the karst wetland of the Yunnan-Guizhou Plateau, Front Environ Sci, 10, https://doi.org/10.3389/fenvs.2022.973900, 2022.

Wu, D., Deng, L., Liu, Y., Xi, D., Zou, H., Wang, R., Sha, Z., Pan, Y., Hou, L., and Liu, M.: Comparisons of the effects of different drying methods on soil nitrogen fractions: Insights into emissions of reactive nitrogen gases (HONO and NO), Atmospheric and Oceanic Science Letters, 13, 224–231, https://doi.org/10.1080/16742834.2020.1733388, 2020.

Zander, F., Heimovaara, T., and Gebert, J.: Spatial variability of organic matter degradability in tidal Elbe sediments, J Soils Sediments, 20, 2573–2587, https://doi.org/10.1007/s11368-020-02569-4, 2020.

Zander, F., Groengroeft, A., Eschenbach, A., Heimovaara, T. J., and Gebert, J.: Organic matter pools in sediments of the tidal Elbe river, Limnologica, 96, 125997, https://doi.org/10.1016/j.limno.2022.125997, 2022.

---

## Author Comment (AC4)

**We thank Reviewer #1 for taking time to give their thorough feedback and useful comments on our manuscript. The text in red indicates our response and the proposed modifications to the manuscript.**

**Reviewer #1:**

The manuscript focuses on organic matter degradation rates in the Amsterdam harbor estuary according to strong anthropogenic influence through dredging activities, using spatial monitoring data and diverse incubation processes. The research is very interesting and meaningful for carbon cycle and greenhouse gas emission from the sediment in such impacted area. The topic of this manuscript fits well with the journal's scope, and the data collected highlighted a strong sampling effort and figures are of good quality.

However, it is important to address some issues in the manuscript before acceptance for publishment, here are some specific comments:

We sincerely thank the reviewer for their careful and constructive feedback during the first review of our manuscript.

Line 80-82: more details in which way shifting salinity affected CH4 are needed (even if discussed in discussion, see comment "line 675-677").

We understand the importance of explaining how shifts in salinity may affect $CH_4$, but in contrast Reviewer #2 suggests to avoid highlighting $CH_4$ here for a smoother transition. We propose deleting here "*particularly for CH4*" to keep the transition more general. We will explain the salinity impact on $CH_4$ in the following sentence. The proposed revised text:

*This can lead to a strong spatial variability in OM degradation pathways and carbon dynamics, particularly for CH4 (Cao et al., 2021). For example, $SO_4^{2-}$ (which covaries with salinity) inhibits $CH_4$ release via multiple mechanisms (e.g. anaerobic oxidation of $CH_4$; Lovley et al., 1987; Egger et al., 2018).*

Line 87: reference is missing.

Line 606: no () for the reference. Dauwe et al. (2001)

Line 642: Zander et al. (2022)

Corrections will be made in the revised manuscript.

Line 89-92: yes, but give examples of naturally and anthropogenically induced sediment disturbance.

We will add examples in the revised manuscript.

Line 111: avoid terms like "our". Here use "the". To be corrected throughout the manuscript (e. g. "our study" replaced by "the present study"…).

Line 139: "our"?

We will change this throughout the revised version of the manuscript.

Line 122-123: out of context here…

This is also related to the next comment raised by Reviewer #1. We propose to make the following modifications to this paragraph:

*Understanding the processing of OM within estuaries takes on further importance because estuarine systems are often intensively altered by human activities (Arndt et al., 2013; Heckbert et al., 2012; Holligan and Reiners, 1992). To increase or maintain waterway navigability, dredging is  commonly practiced in many coastal regions and rivers worldwide. More than 600 million m³ of dredged material is generated annually in Western Europe, China, and the USA (Amar et al., 2021).  While the dredged sediments are often treated as waste and disposed at sea, there is a growing trend of reusing dredged sediments on land, such as beach nourishment, habitat restoration, and land reclamation (Brils et al., 2014), aiming to valorize these massive amounts of dredged materials. However, perturbations during dredging and the following management of the dredged sediments pose an important unknow in the fate of the organic carbon stored in these sediments, with oxygen exposure potentially leading to enhanced carbon remineralization (LaRowe et al., 2020). Given the need for sediment dredging and sustainable management of these materials (van de Velde et al., 2018), it is of great importance to understand to what extent anthropogenic sediment perturbations can affect OM processing and carbon emissions from estuarine sediments.*

Line 138-139: representative of dredged sediment conditioned on land? Is this conditioning the major process for dredged sediment? Line 151 mentioned sediment relocated in the sea. More explanations are needed to justify the choice of these open-air incubations.
We appreciate the reviewer's comments regarding the rationale of the open-air incubations. We will add detailed explanations to justify the choice of open-air incubations in the revised manuscript.

Indeed, as mentioned in Line151, most of the dredged sediments are currently relocated to the North Sea, and accordingly whole-core incubations were performed under water. However, these long-term incubation experiments with submerged sediments (including slurries) may suffer from limited oxygen supply, potentially introducing artefacts in estimating degradation rates. Open-air incubations with optimally wetted sediments, however, avoid such limitations, allowing us to better quantify the oxic degradation potential.

Another reason for performing open-air incubation is linked to the previous comment of Reviewer #1 (see above). Sediment is increasingly reused in (subaerial) applications rather than disposed at sea. Although the open-air incubations in this study are a simplification of on-land applications, findings are still insightful for estimating the carbon emission potential of dredged sediments when exposed to oxygen on land or in oxygenated aquatic environments.

Line 151-152: The rates shown are per year?
Yes.

Line 155-156: how many cores replicates for each location?

Bulk sediments were collected using a single sediment core (Line 155–156), while triplicate cores were used for the incubation of intact sediments (Line 241). Clarification will be added.

Line 242-243: Even evident for the author, precise the reason to have 20 cm of overlying water.

In whole-core sediment incubations, the top 10-20 cm is commonly used because this is the primary zone of diagenesis; an equivalent volume is overlying water is used to (1) avoid rapid depletion/accumulation of dissolved reactants or reactions product to the extent that it affects biogeochemical processes and (2) avoid a large percentage of water exchange (here, 30 mL water, equivalent to 2.3% of overlaying water) when taking discrete samples. We will add an explanation to the revised manuscript.

Line 285: Why 37 days incubation period were chosen?

Thanks for pointing this out. The duration for the incubation was based on the progress of the incubation experiments, as observed in the carbon turnover. The carbon emission rates on day 23, 30 and 37 were much smaller and stable compared to the rates measured early during the experiments (Fig. 6). These data points allow us to extrapolate degradable TOC percentages reliably from 30 days onwards, with 37 days being used as extra check (Fig. 7). The slow carbon from 30 days onward is likely associated with the slow-carbon pool, which is relatively stable. A prolonged incubation is unlikely to substantially alter data interpretation. An explanation will be added.

Line 581: Short range of these different values is needed.

This will be added.

Line 675-677: include the concept of SMTZ (sulfate-methane transition zone)

We will include the SMTZ concept in the revised manuscript.

Line 681: "depending on"

Correction will be made.

Line 687: I don't know if the conclusion needs to be so precise about the results of the study…

Thank you for pointing this out. We will revise the conclusion by removing unnecessary details.

Additional comments: where are the dredging locations on Fig. 1? Dredging affects the upstream and downstream areas of the estuary in a similar way?

Dredging is performed throughout the port area indicated in Fig. 1, albeit with more dredging activities at the marine side compared to the riverine side. This information will be added to section 2.1.

Please give some details on this subject. What would be the consequences of such an imbalance, according to quantity of sediment dredged in each zone?

We acknowledge the reviewer for mentioning this point. We did not yet explore the impact of imbalanced dredging activities across the harbor on sediment OM characteristics and the corresponding implications on carbon release. We will include this to the discussion in the revised manuscript. More dredging activities at the marine site (115) might contribute to the generally lower TOC content (Fig. 2a). However, higher fractions of labile OM (Fig. 7) and faster degradation rates (Fig. 5, 6) in marine sediments suggest that the OM quality/degradability in the investigated area is mainly controlled by the source of the OM and rapid sedimentation. Dredging, although likely enhancing degradation of labile OM, is hence not controlling OM quality/degradability.

What about the sediment relocated in the sea in terms of potential carbon mineralization processes?

We appreciate this consideration. In the Port of Rotterdam (PoR), the current practice of relocating dredged sediment to the North Sea may enhance OM mineralization rate due to enhanced exposure to oxygenated seawater during dredging and redeposition. However, the degree of remineralization is affected by the sediment dynamics after disposal (e.g. sediment resuspension might result in more extensive mineralization, while rapid sediment burial, which limits oxygen exposure, likely enhances organic carbon preservation). We will discuss these aspects in the revised manuscript.

And finally CO2/CH4 effluxes toward the atmosphere?

Ultimately the effluxes to the atmosphere is what matters and we hence appreciated this comment. However, gas exchange between the sea surface and the atmosphere is a research subject in itself and beyond the scope of this manuscript. We will briefly mention this in the revised manuscript as a possible direction for future research.

**References**

Aller, R. C., Blair, N. E., Xia, Q., and Rude, P. D.: Remineralization rates, recycling, and storage of carbon in Amazon shelf sediments, Cont Shelf Res, 16, 753–786, https://doi.org/10.1016/0278-4343(95)00046-1, 1996.

Amar, M., Benzerzour, M., Kleib, J., and Abriak, N. E.: From dredged sediment to supplementary cementitious material: characterization, treatment, and reuse, International Journal of Sediment Research, 36, 92–109, https://doi.org/10.1016/j.ijsrc.2020.06.002, 2021.

Arndt, S., Jørgensen, B. B., LaRowe, D. E., Middelburg, J. J., Pancost, R. D., and Regnier, P.: Quantifying the degradation of organic matter in marine sediments: A review and synthesis, Earth Sci Rev, 123, 53–86, https://doi.org/10.1016/j.earscirev.2013.02.008, 2013.

Brils, J., de Boer, P., Mulder, J., and de Boer, E.: Reuse of dredged material as a way to tackle societal challenges, J Soils Sediments, 14, 1638–1641, https://doi.org/10.1007/s11368-014-0918-0, 2014.

Cao, C., Cai, F., Qi, H., Zhao, S., and Wu, C.: Differences in the sulfate–methane transitional zone in coastal pockmarks in various sedimentary environments, Water (Switzerland), 13, 1–18, https://doi.org/10.3390/w13010068, 2021.

Heckbert, S., Costanza, R., Poloczanska, E. S., and Richardson, A. J.: Climate Regulation as a Service from Estuarine and Coastal Ecosystems, Elsevier Inc., 199–216 pp., https://doi.org/10.1016/B978-0-12-374711-2.01211-0, 2012.

Holligan, P. M. and Reiners, W. A.: Predicting the Responses of the Coastal Zone to Global Change, Adv Ecol Res, 22, 211–255, https://doi.org/10.1016/S0065-2504(08)60137-3, 1992.

LaRowe, D. E., Arndt, S., Bradley, J. A., Estes, E. R., Hoarfrost, A., Lang, S. Q., Lloyd, K. G., Mahmoudi, N., Orsi, W. D., Shah Walter, S. R., Steen, A. D., and Zhao, R.: The fate of organic carbon in marine sediments - New insights from recent data and analysis, Earth Sci Rev, 204, 103146, https://doi.org/10.1016/j.earscirev.2020.103146, 2020.

van de Velde, S., van Lancker, V., Hidalgo-Martinez, S., Berelson, W. M., and Meysman, F. J. R.: Anthropogenic disturbance keeps the coastal seafloor biogeochemistry in a transient state, Sci Rep, 8, 1–10, https://doi.org/10.1038/s41598-018-23925-y, 2018.

---

## Author Comment (AC5)

**We thank Reviewer #2 for the detailed and constructive feedback. The text in red indicates our response and the proposed modifications to the manuscript.**

**Reviewer #2:**

"Perturbation increases source-dependent organic matter degradation rates in estuarine sediments" by Guangnan Wu et al. is a comprehensive and well-executed study investigating the chemical characteristics of sediment organic matter in Rotterdam harbour and the potential response of sediment carbon stocks to human disturbances (i.e. dredging). The study employs a range of techniques in the basic characterization of samples along a salinity transect (bulk sediment C/N, del13C, BIT index and pyrolysis-GCMS analysis of macromolecular organic matter) followed by a set of incubation experiments (both classical whole-core and homogenized sediment subaerial set-ups) to show how remineralization processes respond to disturbance. The data is of high quality and is generally processed and interpreted well, with a high degree of integration between the various lines of evidence. Overall, the outcome is convincing and should be published. However, I outline below one key issue related to the data processing that requires more careful consideration in the text, and several minor comments that should be addressed.

We sincerely thank the reviewer for their positive recommendation as well as the constructive feedback on our manuscript.

Key issue:

In processing the results of the whole-core incubations, ammonium fluxes are used to estimate the fraction of the DIC flux that is derived from organic matter remineralization (results shown in Fig. 5e). This calculation assumes Redfield stoichiometry for the degrading material, i.e. ammonium and DIC from remineralization are released in the ratio 106:16. This may be valid for the marine end-member site 115 but not necessarily for riverine end-member site 21A. Considering the bulk sediment C/N ratio data it is likely that site 21A releases more DIC per mole ammonium, even if the most reactive fraction of the sedimentary organic matter is relatively nitrogen rich. I suggest that this uncertainty is somehow included in the calculations, for example by presenting additional bars in Fig. 5e or a summary table. This issue is important because the quantification of remineralization-derived DIC production in the whole-core incubations is used later in the study when comparing with rates in the subaerial incubations, ie. assessing by how much dredging of sediment onto land would stimulate remineralization.

We appreciate the reviewer for pointing this out. We propose to change Fig 5e accordingly (see below). This way we include impact of the differences in the organic matter (OM) composition and C/N ratio between site 115 and 21A (Table 2, see below). We will use different C/N ratios (with literature-reported ranges, Table 2) to account for differences in OM composition between sites 115 and 21A and the uncertainties associated with this will also be presented in a summary table (as Table 2).

However, we want to mention as well that the bulk sediment C/N ratio (Table 2) does not necessarily reflect the stoichiometry of the degraded OM during incubation, which is often the most labile fraction.

The most labile OM, which is preferentially mineralized during incubation, is typically more nitrogen-rich, particularly in systems with mixed OM sources (Albert et al., 2021; Arndt et al., 2013). Therefore, we expect that the actual C/N ratio of the degraded fraction likely falls towards the lower ends of the literature-based ranges (Table 2), and the DIN-corrected DIC fluxes are more likely to be closer to the "minimum" values in Figure 5e and Table 2.

We will revise the manuscript accordingly to clarify these points and to explain the rationale behind our calculations. Proposed new Fig. 5, supplemented by Table 2:

[Figure]

**Fig. 5.** Benthic fluxes of dissolved $O_2$ (a), DIC (b), and $CH_4$ (c) determined from whole-core incubation. Positive and negative rates represent efflux (from sediment into overlying water) and influx (from overlying water into sediment), respectively. Sediment TOC-normalized DIC (DIC norm) is presented in panel (d) with TOC content being 2.2 wt.% for 115 and 5.0 wt.% for 21A. Panel (e) shows DIN-corrected DIC fluxes (representing OM-derived DIC), normalized by sediment TOC (corr DIC norm), using minimum (min) and maximum (max) C:N ratios based on literature (see Table 2). Error bars represent the standard deviation from triplicate core incubations.

**Table 2.** The bulk C/N ratios at sites 115 and 21A compared to values reported in literature in marine and riverine sediments. The literature-reported ratios were used to calculate DIN-corrected DIC fluxes (Fig. S4c) and TOC-normalized, DIN-corrected DIC fluxes (Fig. 5e) for sites 115 and 21A.

| Sediment | Site 115 (marine) | Site 21A (riverine) |
|---|---|---|
| Measured bulk C:N ratio | 7.2 | 9.9 |
| Literature-reported bulk C:N ratio in marine/riverine sediments | 6.625 (Redfield, 1958), 7.4 (Martiny et al., 2014), 5.6–7.5 (Körtzinger et al., 2001) | 12 (Benoit et al., 2006), 13.1 (They et al., 2017), 7.1–13 (Maranger et al., 2018), 10.4–14.2 (Hecky et al., 1993) |
| Measured DIN flux | $9.8 \pm 1.3$ mmol m$^{-2}$ d$^{-1}$ | $7.0 \pm 0.9$ mmol m$^{-2}$ d$^{-1}$ |
| Min C:N ratio for conversion | 5.6 | 7.1 |
| Max C:N ratio for conversion | 7.5 | 14.2 |
| DIN-corrected DIC flux (Fig. S4c), min mean (± SD)–max mean (± SD) | 54.7 (±7.2)–73.3 (±9.7) mmol m$^{-2}$ d$^{-1}$ | 49.8 (±6.5)–99.7 (±12.9) mmol m$^{-2}$ d$^{-1}$ |
| DIN-corrected DIC flux with TOC normalization (Fig. 5e), min mean (± SD)–max mean (± SD) | 62.2 (±8.2)–83.2 (±11.0) µg C g$^{-1}$ day$^{-1}$ | 17.6 (±1.0)–35.1 (±2.0) µg C g$^{-1}$ day$^{-1}$ |

Minor comments:

Line 81-82: No need to highlight CH4 specifically. The point is valid for the balance between any given set of remineralization pathways.

We agree there is no need to highlight $CH_4$ specifically in this sentence, but it is important to mention somewhere that changes in salinity affect $CH_4$ dynamics (as also mentioned by the Reviewer #1). Accordingly, we propose to delete "particularly for $CH_4$" in this sentence, keeping the transition in the

*text more general (Line 81–82), and introduce salinity-CH$_4$ dynamics in the following sentence. The revision will as follows:*

> *This can lead to a strong spatial variability in OM degradation pathways and carbon dynamics, particularly for CH$_4$ (Cao et al., 2021). For example, SO$_4^{2-}$ (which covaries with salinity) inhibits CH$_4$ release via multiple mechanisms (e.g. anaerobic oxidation of CH$_4$; Lovley et al., 1987; Egger et al., 2018).*

Line 84-87: Consider rewording the sentence ("estuaries" appears to be used in both a general and a specific way in the same sentence) and add the reference.

*Will be changed accordingly.*

Line 88: Should be "penetration depth".

Line 99: Make sure to use brackets correctly.

Line 123: Should be "dredged sediment".

Line 137-139: Should be "bottle incubations"

Line 139: Should be "Our results show..." or "We show..."

Line 141: Should be "the properties of OM influence..."

Line 213: Remove "an"

Line 383: Should be "overlying water".

Line 438: Should be "thus" not "this"

Line 478: Should be "may explain the observation that".

Line 607: Check the formatting of the citation.

Line 620: Should be "10 times faster than..."

*We will double check and correct the typos and grammar throughout the manuscript.*

Line 162: Use "end-members" or similar, rather than "realm"

*Will be changed.*

Line 220-229: It would be useful for a reader trying to reproduce this method to know how the HF was disposed of during the protocol. Is it evaporated at some stage? This is a dangerous chemical and lab protocols need to be carefully designed.

*The HF waste was collected in a high-density polyethylene plastic container designated for HF waste, further handled by the EHS team (i.e. ARBO staff in the Netherlands). Details will be added in the revised manuscript.*

Line 229: Clarify that "MOM" here refers to the residue of the previous steps.

*Clarification will be made.*

Line 283: It is not clear what is meant by "60% water-filled volume". It is difficult to estimate the porosity of freeze-dried and homogenized sediment after re-wetting due to changes in grain size distribution during these processing steps.

We thank the reviewer for the comment. Here, the "60% water-filled pore space" refers to 60% of the porosity of the freeze-dried, homogenized sediment. We agree that the sediment structure may change after rewetting, which could further affect the sediment porosity. To avoid confusion, we propose to simply describe the materials as ''rewetted sediments". For clarity, we will specify that the added volume of artificial rainwater corresponds to 60% of the porosity of the freeze-dried sediment.

Line 345: "Asterisk/stars" not "asteroids". Also unclear what the arrows in the plot indicate, should they be pointed in the opposite direction to link the numerical labels to the 2D fields in the x-y plot?
Will be changed accordingly.

Line 245: Should be Table S3, not S2.
We will double check ordering of tables and figures.

Fig. 6: Please add the information about "marine" vs. "riverine" stations to the plot or legend.
Will be added.

Fig. 7: What do the stipple vs. solid lines indicate? Also show which sites are "marine" and "riverine".
Different line styles were intended to differentiate two sediment textures (i.e. sand-rich sediment vs. silt-rich sediment). To avoid confusion, we will use the solid lines for all subplots in Fig. 7. We will also specify in the new plot that site 115 is the "marine" site, and site 21A is the "riverine" site.

Line 451: This is a valid consideration rather than a confounding factor.
Will be changed.

Line 517: It is not unusual that soil OM contains similar chemical signatures to the higher plants that grow and decay in them. It is perhaps misleading to state that the correlation between BIT and the EMMM results implies low contribution of vegetation input, rather it should be stated that, as you conclude later, the plant signals are transferred via the soils (where they also pick up the BIT signal).
Thank you for noticing. The text will be revised as follows:

> *Its strong correlation with the modelled non-marine OM (encompassing soil OM, riverine OM, and terrestrial vegetation input) suggests that vegetation input was not a major component of the modelled non-marine OM contribution. , or if not, vegetation derived OM was transferred via the soils, where it became associated with an (enhanced) BIT signal. The latter assumption was supported by the MOM constituents, where the relative abundance of lignin-derived products (indicating higher plant signals) was up to 40% (Fig. 8b).*

Line 551-552: The comparison with ocean margin trends is not useful here as the reactivity gradients in near-shore areas is more likely to be controlled by terrestrial inputs than by water-depth fractionation of sinking OM.

We agree with the reviewer and this comparison will be removed from the revised manuscript.

Line 564: Remove "ancient". There is no information about age of the material so this is not relevant.

Will be removed.

Line 567-568: Interpretation of carbonate dissolution needs to be expanded. Is there evidence for this in the data or literature?

Yes, the DIC flux/DIN flux was ~16, which exceeds the literature-based C/N ratio (5.6–14.2; Table 2),and is much larger than the measured bulk C/N ratios (7.2 and 9.9; Table 2). This implies that other processes, besides OM mineralization, must contribute to the carbon release. When $CO_2$ is added to porewater carbonate dissolution is likely to occur. We propose the following changes in the revised text to make this more clear.

*The respiratory quotient (RQ), determined as the ratio between DIC outflux and $O_2$ influx, was notably higher in the investigated estuarine sediments (3.75–5) than the typical range observed in marine sediments (0.69–1.31; Jørgensen et al., 2022), probably because carbonate dissolution enhances the DIC flux suggested by the DIC flux to DIN flux ratios (~16; Fig S4b). The number exceeded the C/N ratios reported in the literature (5.6–14.2; Table 2) and the measured bulk C/N ratios (7.2 and 9.9; Table 2). Notably, fresh, nitrogen-rich OM with lower C/N ratios is usually preferentially degraded first during incubation.*

Line 577: The higher CH4 flux in a whole core incubation at a more freshwater site can be due to the higher position in the sediment column of the SMTZ. See e.g. the diagram in Fig. 7 of https://esd.copernicus.org/articles/13/633/2022/esd-13-633-2022.html.

Modification will be made.

Line 633: Also refer to Keil et al. (1994).

Reference will be added.

Line 677-678: Separate the statements about global AOM estimates and dredging impacts, these are not obviously connected.

We propose the following revisions to the text in this paragraph:

*Methane, a strong greenhouse gas, is often oversaturated in the OM-rich coastal sediments where $CH_4$ bubbles are formed. Most $CH_4$ is trapped below the sulfate-methane transition (SMTZ), and approximately 71% of the $CH_4$ is ultimately removed via anaerobic oxidation of methane (AOM) in marine sediments (Gao et al., 2022). However, the removal efficiency is usually lower in estuarine sediments due to lower $SO_4^{2-}$ concentrations. Dredging, which can remove the SMTZ completely or partially, is likely to impact the functioning of the AOM filter, consequently leading to a short-term $CH_4$ emission peak by increasing diffusion and ebullition (Maeck et al., 2013; Nijman et al., 2022). Estuarine systems are characterized by a strong salinity gradient with a large variability of the depth of the sedimentary methanic zone. Anaerobic oxidation*

*of methane consumes approximately 71% of the CH₄ in marine sediments (Gao et al., 2022), while dredging will inevitably disrupt anaerobic methane oxidation. While dredging intensity and the SMTZ position can differ greatly within estuaries, further research should quantify the effect of dredging on CH₄ emission considering realistic, large scale and site-specific dredging practices.*

**References**

Albert, S., Bonaglia, S., Stjärnkvist, N., Winder, M., Thamdrup, B., and Nascimento, F. J. A.: Influence of settling organic matter quantity and quality on benthic nitrogen cycling, Limnol Oceanogr, 66, 1882–1895, https://doi.org/10.1002/lno.11730, 2021.

Arndt, S., Jørgensen, B. B., LaRowe, D. E., Middelburg, J. J., Pancost, R. D., and Regnier, P.: Quantifying the degradation of organic matter in marine sediments: A review and synthesis, Earth Sci Rev, 123, 53–86, https://doi.org/10.1016/j.earscirev.2013.02.008, 2013.

Benoit, P., Gratton, Y., and Mucci, A.: Modeling of dissolved oxygen levels in the bottom waters of the Lower St. Lawrence Estuary: Coupling of benthic and pelagic processes, Mar Chem, 102, 13–32, https://doi.org/10.1016/j.marchem.2005.09.015, 2006.

Cao, C., Cai, F., Qi, H., Zhao, S., and Wu, C.: Differences in the sulfate–methane transitional zone in coastal pockmarks in various sedimentary environments, Water (Switzerland), 13, 1–18, https://doi.org/10.3390/w13010068, 2021.

Gao, Y., Wang, Y., Lee, H. S., and Jin, P.: Significance of anaerobic oxidation of methane (AOM) in mitigating methane emission from major natural and anthropogenic sources: a review of AOM rates in recent publications, Environmental Science: Advances, 1, 401–425, https://doi.org/10.1039/d2va00091a, 2022.

Hecky, R. E., Campbell, P., and Hendzel, L. L.: The stoichiometry of carbon, nitrogen, and phosphorus in particulate matter of lakes and oceans, Limnol Oceanogr, 38, 709–724, https://doi.org/10.4319/lo.1993.38.4.0709, 1993.

Jørgensen, B. B., Wenzhöfer, F., Egger, M., and Glud, R. N.: Sediment oxygen consumption: Role in the global marine carbon cycle, Earth Sci Rev, 228, https://doi.org/10.1016/j.earscirev.2022.103987, 2022.

Körtzinger, A., Hedges, J. I., and Quay, P. D.: Redfield ratios revisited: Removing the biasing effect of anthropogenic CO2, Limnol Oceanogr, 46, 964–970, https://doi.org/10.4319/lo.2001.46.4.0964, 2001.

Maeck, A., Delsontro, T., McGinnis, D. F., Fischer, H., Flury, S., Schmidt, M., Fietzek, P., and Lorke, A.: Sediment trapping by dams creates methane emission hot spots, United States of America, Environmental Science and Technology, 8130–8137, 2013.

Maranger, R., Jones, S. E., and Cotner, J. B.: Stoichiometry of carbon, nitrogen, and phosphorus through the freshwater pipe, Limnol Oceanogr Lett, 3, 89–101, https://doi.org/10.1002/lol2.10080, 2018.

Martiny, A. C., Vrugt, J. A., and Lomas, M. W.: Concentrations and ratios of particulate organic carbon, nitrogen, and phosphorus in the global ocean, Sci Data, 1, https://doi.org/10.1038/sdata.2014.48, 2014.

Nijman, T. P. A., Lemmens, M., Lurling, M., Kosten, S., Welte, C., and Veraart, A. J.: Phosphorus control and dredging decrease methane emissions from shallow lakes, Science of the Total Environment, 847, 157584, https://doi.org/10.1016/j.scitotenv.2022.157584, 2022.

Redfield, A. C.: THE BIOLOGICAL CONTROL OF CHEMICAL FACTORS IN THE ENVIRONMENT, 205–221 pp., 1958.

They, N. H., Amado, A. M., and Cotner, J. B.: Redfield ratios in inland waters: higher biological control of C:N:P ratios in tropical semi-arid high water residence time lakes, Front Microbiol, 8, https://doi.org/10.3389/fmicb.2017.01505, 2017.

---

## Author Comment (AC6)

**We thank Reviewer #3 for taking time to give their feedback. The text in red indicates our response and the proposed modifications to the manuscript.**

**Reviewer #3:**

The manuscript of G. Wu et al. investigates the increase of organic matter (OM) degradation produced by anthropogenic perturbations such as dredging activity. The method stands on the elemental (C/N) and isotopic ($\delta$13C) characterization of organic matter of 49 sites from the Port of Rotterdam. Molecular characterization was done on 13 of the 49 sites, reoxydation experiments on 6 sites and in situ OM remineralization have been estimated on 2 sites. The authors conclude from their studies that Cl#1 - CN proxies are robust to identify contribution of marine versus non-marine OM sources, Cl#2 - marine OM is more labile than terrestrial OM and Cl#3-aerobic conditions accelerates OM remineralization compared to natural (mainly anaerobic) conditions.

From my point of view, this manuscript is not reaching the quality requirement to get published in BG : the rare new knowledges brought by this manuscript are only site specific. In the conclusion, the authors underline more general concepts (Cl#1, Cl#2, Cl#3), however it corresponds to already well-established concept in biogeochemistry that does not present significant novelty.

We thank the reviewer for their time and effort in evaluating our manuscript. Reviewer #3's assessment about the quality and significance of the work strongly contrasts with evaluations by Reviewer #1 and #2, who highlight the importance and high quality of the work. Uniquely, our work combines detailed sediment and OM characterization with incubation of intact and disturbed sediments to link $CO_2$ emission potential to OM properties (rather than content) and biogeochemical processes as function of environmental conditions. Whereas some of the broader concepts discussed (e.g. selective preservation, aerobic mineralization) are indeed established in biogeochemistry, the contribution of this study lies in applying and contextualizing these concepts within the complex settings of a heavily perturbed estuarine system, which is understudied in existing literature. In addition, the findings provide crucial insights into how human-induced disturbances interact with natural OM processing, with implications for carbon cycling. Such knowledge is essential for informed policy making on processing dredging waste. As estuarine systems face increasing pressure of human perturbation, we believe this study represents a meaningful contribution to the field, as also noted by both Reviewer #1 and #2. Where possible, we will emphasize the significance of this new knowledge in the abstract, introduction and discussion of the manuscript.

In addition, a detailed reading of the manuscript reveal that the scientific hypothesis defended along the manuscript (that perturbation increases source-dependent organic matter degradation rates in estuarine sediment) is not properly argued. While the source characterization of the OM received large analytical efforts, and a valid interpretation supported by some modelisation, four others critical step of the demonstration seems not strong enough. First, and most important, the demonstration of a source and composition dependence of OM degradation rate is not convincing. The authors build their

hypothesis from the literature and a reasonable intuition but that is in complete opposition with the dataset presented. Indeed, the two sites, selected because they present very contrasted OM origin and composition, present very similar O2 fluxes that is interpreted -reasonably- as a similar intensity of remineralization rate. This result should better be used to underline a limit of the observations from the literature. Instead, the authors use an artefact (normalization by the TOC) to build a contrast between the two stations while the relation between quantity of OM and remineralization rate have not been presented. It seems more rational to conclude from their dataset that OM quantity does not play an important role in deposit, probably because it is in large excess.

We agree that sedimentary organic carbon turnover might to some extent depend on $O_2$ availability, but we want to clarify that the $O_2$ fluxes represent the oxygen consumption by remineralization of fresh OM deposited at the sediment surface, which is only a small fraction of the total organic matter pool present. When considering the entire sediment succession (or sediment cores) the oxygen penetration depth is usually a few millimeters only or even less in OM-rich coastal sediments (Cai and Sayles, 1996). We will explain this argument more carefully in the revised manuscript.

We do not agree with the suggestion that using TOC-normalized rates introduces an artefact. Normalization on bulk sedimentary TOC is commonly used to assess the relative reactivity of OM (e.g. Freitas et al., 2025; Zander et al., 2020, 2022), especially when TOC content differs considerably between sites (in this case 2.2 wt.% and 5 wt.%). The TOC-normalization allows comparing OM degradation rates between sites with different OM compositions (more fresh algal material at site 115 compared to the more refractory terrestrial-derived OM at site 21A) and, furthermore, between different incubation conditions (Fig. 5e and 6b). Importantly, it highlights that caution is required when interpreting bulk OM content in terms of $CO_2$ release potential.

Second in importance, there is no consideration on primary production, while freshly produced algea are probably more labile than any other carbon source. In other word the local recycling of carbon between photosynthesis and respiration is neglected in this manuscript while it is probably very important in this very shallow environment.

We agree with the referee, and this is actually one of the points we tried to make in the manuscript, however, apparently not clearly enough. Primary production and the associated input of labile algal OM, which fuels the remineralization are important aspects of the manuscript. We first mentioned it in the whole-core incubation experiments, where the local OM production and transport dominate (Line 561–565). Then we expanded this aspect to the discussion of carbon release from mixed, dredged sediments (Line 619–621). We later mentioned it again in the conclusion section (Line 704–710). We will try to connect these sections with an overarching statement to address the concern of this reviewer.

Third, while the introduction suggests a study covering many sites and embracing some spatial variability, the comparison between perturbated and not perturbated sediment is done only on 2

sediment, which seems not enough to support conclusions that could be generalized to others estuaries, and thus limits the interest of the results.

Within the scope of this project, we could only perform whole-core incubations at a small number of sites; we therefore selected strongly contrasting sites to show relationships between OM composition, degradation and $CO_2$ emission. These parameters will always be strongly affected by local conditions (e.g, vegetation, hydrodynamics, temperatures), especially for highly dynamic estuaries at the land-sea interface. Our work does likely reflect general relationships between depositional environment (marine vs. riverine), OM composition and $CO_2$ release potential in human-impacted estuaries (we will highlight this better in the Implications section).

Fourth, the methodology of perturbation does not seem adapted to an estuarine environment, since it corresponds to a not water-saturated sediment more adapted to mimic soil aeration than resuspension induced by dredging activities.

The choice of experiments was based on mimicking natural conditions only, but also includes deliberate perturbations, as was noted by Reviewer #1 as well. We will add a detailed explanation in both Introduction and Discussion sections of the revised manuscript. In short, the open-air incubation was chosen because: (i) the submerged sediment incubation (e.g. slurry experiment) may suffer from $O_2$ supply, potentially introducing artefacts in estimating degradation rates. Open-air incubations with optimally wetted sediments, however, avoid such limitations, allowing us to better quantify the oxic degradation potential; and (ii) open-air incubation tests the impact of on-land application (which is an increasing practice in sediment management) to carbon emission.

Additionally, the dry freezing of the sediment before incubation would certainly modify the properties of the reactive organic matter – which have not been tested.

We acknowledge that freeze-drying may have changed reactive OM properties. However, using alternative methods like air-drying likely also changes OM characteristics and at the same time allow OM remineralization before the experiment as air-drying usually lasts a few days. Using freeze-drying is hence maybe not ideal, but seems here the best possible option, also as earlier incubation studies using freeze drying reported limited impact of freeze-drying on carbon emissions (He et al., 2022; Wu et al., 2020). Fromin (2025) suggests that there is rarely consensus on a best practice when studying microbial processes in soils and sediments, and the appropriate methods often depend on the specific goals of the study. Given the limitations of different approaches and the aim to quantify the labile OM fraction, freeze-drying was considered the more suitable option for sample pre-treatment in our case.

**References**

Cai, W.-J. and Sayles, F. L.: Oxygen penetration depths and fluxes in marine sediments, Mar Chem, 52, 123–131, https://doi.org/https://doi.org/10.1016/0304-4203(95)00081-X, 1996.

Freitas, N. L., Walter Anthony, K., Lenz, J., Porras, R. C., and Torn, M. S.: Substantial and overlooked greenhouse gas emissions from deep Arctic lake sediment, Nat Geosci, https://doi.org/10.1038/s41561-024-01614-y, 2025.

Fromin, N.: Impacts of soil storage on microbial parameters, SOIL, 11, 247–265, https://doi.org/10.5194/soil-11-247-2025, 2025.

He, Y., Zhang, T., Zhao, Q., Gao, X., He, T., and Yang, S.: Response of GHG emissions to interactions of temperature and drying in the karst wetland of the Yunnan-Guizhou Plateau, Front Environ Sci, 10, https://doi.org/10.3389/fenvs.2022.973900, 2022.

Wu, D., Deng, L., Liu, Y., Xi, D., Zou, H., Wang, R., Sha, Z., Pan, Y., Hou, L., and Liu, M.: Comparisons of the effects of different drying methods on soil nitrogen fractions: Insights into emissions of reactive nitrogen gases (HONO and NO), Atmospheric and Oceanic Science Letters, 13, 224–231, https://doi.org/10.1080/16742834.2020.1733388, 2020.

Zander, F., Heimovaara, T., and Gebert, J.: Spatial variability of organic matter degradability in tidal Elbe sediments, J Soils Sediments, 20, 2573–2587, https://doi.org/10.1007/s11368-020-02569-4, 2020.

Zander, F., Groengroeft, A., Eschenbach, A., Heimovaara, T. J., and Gebert, J.: Organic matter pools in sediments of the tidal Elbe river, Limnologica, 96, 125997, https://doi.org/10.1016/j.limno.2022.125997, 2022.

---

## Author Response (AR1)

Dear Editor and Reviewers,

Thank you for taking time to review our manuscript titled '*Perturbation increases source-dependent organic matter degradation rates in estuarine sediments*". Below we provide detailed responses to reviewers' comments and elaborate how each point has been incorporated into the revised manuscript.

Kind regards,
Guangnan Wu, on behalf of the coauthors

NB: Our response is given in red, also referring to the line numbers in the revised manuscript with Word's **simple markup** where the edited text can be found. Quotes of the newly edited text are indicated with "quotation marks". For ease, we mention the topic of a major comment at the start [*between parentheses in italic*] and use these when referring to other responses within our rebuttal.
* * *
**Editor's comments:**

Dear authors,

[*Site selection and method*ology] As you noticed, contrasted reviews were made in terms of significance for this study. You must be very carefully in the new version to clarify the differential analytical and experimental efforts made (13, 6 and 2 sites for different kind of experiments)
We added explanatory sentences at the beginning of sections 2.2–2.6 that describe the rationale behind site selection for the various analytical and experimental efforts:

- Bulk sediment analysis (Section 2.1, 2.2): "Bulk sediments were collected from 49 selected locations throughout the study area in the summer of 2021. These sites were selected from over 300 monitoring sites in the Port of Rotterdam to represent the full spectrum of depositional conditions in the main waterway and adjacent harbor areas from marine to riverine (Fig. 1)." (L159−162)

- Lipids and MOM analysis (Section 2.3, 2.4): "Sediments from 13 key locations (Fig. 1), selected to cover the full river-marine salinity transect, were used for lipids and MOM analyses." (L214−215)

- Subaerial/open-air bottle incubation (Section 2.6): "To investigate OM degradability under oxygen exposure during dredged sediment processing while avoiding oxygen supply as a limitation, open-air bottle incubations were conducted in triplicate for six sediments that covered contrasting depositional and sedimentary conditions within the research area: three marine (115, 86, NWWG-02; Fig. 1a) and three riverine (21A, B16, K1v2; Fig. 1a), with differing sediment texture (silt-rich and sand-rich) in both groups." (L297−301)

- Whole-core incubation (Section 2.5): "Triplicate intact sediment cores collected from two strongly contrasting sites (marine site 115 vs. riverine site 21A) were used for whole-core incubation. These

sites represent relatively intensively dredged marine and riverine areas, respectively, that contribute significantly to the total annual dredged sediment volume in the PoR." (L252−255)

[*O₂ fluxes*] Especially please consider limitations pointed by reviewer 3 on the interpretation of O2 fluxes maybe expanding a bit the discussion on the organic matter excess and also on the potential reoxydation rates and related CO2 fluxes while it comes to less labile organic matter.

We have added more careful consideration of the $O_2$ fluxes, for details please see our response to Reviewer #3's comment about [*O₂ fluxes*] below.

[*Sample processing*] In the material and methods section you should put more clearly, limitations brought by freeze-drying sediment in comparison with other methods and clearly explain why freeze drying is the most suitable in your case.

We now better justify this choice (minimal sediment alteration, reproducibility: L301−303) and discuss its (likely limited) impact (L653−657).

[*Perturbation and CH₄ escape*] If I may suggest some reading, get a look on Hulot et al (2023) and Barhdadi et al (2024) papers dealing with a decennial flood in the Loire and how this generates erosion and methane ebullition affecting benthic fluxes that are dominated by benthic advective exchanges. This is a good example of natural erosion effects that could be extrapolated to dredging effects even if it doesn't deal with the reactivity of suspended matter.

We thank the editor for this suggestion and have carefully considered the articles; we have included Hulot et al.'s work in our discussion of perturbation and $CH_4$ escape (L725−726).

Sincerely
Edouard
* * *
**Reviewer #1:**

The manuscript focuses on organic matter degradation rates in the Amsterdam harbor estuary according to strong anthropogenic influence through dredging activities, using spatial monitoring data and diverse incubation processes. The research is very interesting and meaningful for carbon cycle and greenhouse gas emission from the sediment in such impacted area. The topic of this manuscript fits well with the journal's scope, and the data collected highlighted a strong sampling effort and figures are of good quality.

However, it is important to address some issues in the manuscript before acceptance for publishment, here are some specific comments:

We sincerely thank the reviewer for their careful and constructive feedback during the first review of our manuscript.

[*Salinity and CH₄*] Line 80-82: more details in which way shifting salinity affected CH4 are needed (even if discussed in discussion, see comment "line 675-677").

We have added information about the role of $SO_4^{2-}$ in the microbial sediment $CH_4$ filter (L81−83).

Line 87: reference is missing.

Added (L89).

Line 606: no () for the reference. Dauwe et al. (2001)

Line 642: Zander et al. (2022)

Spelling and grammar were checked and corrected in the revised manuscript.

Line 89-92: yes, but give examples of naturally and anthropogenically induced sediment disturbance.

Added (L92−93).

Line 111: avoid terms like "our". Here use "the". To be corrected throughout the manuscript (e. g. "our study" replaced by "the present study"…).

Line 139: "our"?

Corrections were made throughout the revised manuscript.

Line 122-123: out of context here…

We modified the paragraph to accommodate and highlighted the significance of sediment reuse (L121−127).

[*Subaerial bottle incubation experiment*] Line 138-139: representative of dredged sediment conditioned on land? Is this conditioning the major process for dredged sediment? Line 151 mentioned sediment relocated in the sea. More explanations are needed to justify the choice of these open-air incubations.

In our revision, we emphasized the increased interest in sediment reuse rather than relocation to sea, including subaerial applications (L121−123) and specifically linked the bottle incubation experiment to this (L138-139). In section 4.4, (L710-720), we further elaborated on the implications of subaerial sediment applications.

Line 151-152: The rates shown are per year?

Yes; added (L154).

Line 155-156: how many cores replicates for each location?

Clarification was added. Bulk sediments were collected using a single sediment core (L162), while triplicate cores were used for the incubation of intact sediments (L252).

Line 242-243: Even evident for the author, precise the reason to have 20 cm of overlying water.

We added an explanation to the revised manuscript (L255−259).

Line 285: Why 37 days incubation period were chosen?

Within the scope and constraints of the project, we focused on short-term $CO_2$ emissions and terminated the experiment when rates stabilized at low levels after about one month. This is now explained in L309−311.

Line 581: Short range of these different values is needed.

Added (L610).

Line 675-677: include the concept of SMTZ (sulfate-methane transition zone)

Added (L723).

Line 681: "depending on"

Changed (L730).

Line 687: I don't know if the conclusion needs to be so precise about the results of the study…

We shortened the conclusion by removing unnecessary details in the revised manuscript (L737−750)

Additional comments: where are the dredging locations on Fig. 1? Dredging affects the upstream and downstream areas of the estuary in a similar way?

Dredging is ubiquitous and heterogeneous in the harbor area and not easily captured in a simple map, with largest volumes removed from the more recent and downstream (marine) port area (L151−158). We mention the relatively large contribution from areas around sites 115 and, to a lesser extent, site 21 (L253−255).

Please give some details on this subject. What would be the consequences of such an imbalance, according to quantity of sediment dredged in each zone?

More intensive dredging occurs in the marine area, therefore dredging predominantly perturbs sediment with relatively labile OM (L664−666).

What about the sediment relocated in the sea in terms of potential carbon mineralization processes?

Discussion was added in the revised manuscript, detailing the different fates of sedimentary C under marine and subaerial conditions (L710−720).

And finally CO2/CH4 effluxes toward the atmosphere?

Ultimately the efflux to the atmosphere is what matters and we hence appreciated this comment. However, gas exchange between the sea surface and the atmosphere was beyond the scope of this study. The relationship between our results and implications for estuarine $CO_2$ and $CH_4$ release into the atmosphere are mentioned in L611−614 and L722−729.

**Reviewer #2:**

"Perturbation increases source-dependent organic matter degradation rates in estuarine sediments" by Guangnan Wu et al. is a comprehensive and well-executed study investigating the chemical characteristics of sediment organic matter in Rotterdam harbour and the potential response of sediment carbon stocks to human disturbances (i.e. dredging). The study employs a range of techniques in the basic characterization of samples along a salinity transect (bulk sediment C/N, del13C, BIT index and pyrolysis-GCMS analysis of macromolecular organic matter) followed by a set of incubation experiments (both classical whole-core and homogenized sediment subaerial set-ups) to show how remineralization processes respond to disturbance. The data is of high quality and is generally processed and interpreted well, with a high degree of integration between the various lines of evidence. Overall, the outcome is convincing and should be published. However, I outline below one key issue related to the data processing that requires more careful consideration in the text, and several minor comments that should be addressed.

We sincerely thank the reviewer for their positive recommendation as well as the constructive feedback on our manuscript.

Key issue:

[*Calculation OM-derived DIC fluxes*] In processing the results of the whole-core incubations, ammonium fluxes are used to estimate the fraction of the DIC flux that is derived from organic matter remineralization (results shown in Fig. 5e). This calculation assumes Redfield stocihiometry for the degrading material, i.e. ammonium and DIC from remineralization are released in the ratio 106:16. This may be valid for the marine end-member site 115 but not necessarily for riverine end-member site 21A. Considering the bulk sediment C/N ratio data it is likely that site 21A releases more DIC per mole ammonium, even if the most reactive fraction of the sedimentary organic matter is relatively nitrogen rich. I suggest that this uncertainty is somehow included in the calculations, for example by presenting additional bars in Fig. 5e or a summary table. This issue is important because the quantification of remineralization-derived DIC production in the whole-core incubations is used later in the study when comparing with rates in the subaerial incubations, ie. assessing by how much dredging of sediment onto land would stimulate remineralization.

We thank the reviewer for pointing out the uncertainty regarding the OM-driven DIC flux, which we estimate by using the DIN flux and the Redfield C/N ratio. The uncertainty introduced by assuming Redfield was likely limited: the bulk C/N ratios in the surface sediment were 7.2 (115) and 9.9 (21A)— the somewhat elevated C/N at site 21 will likely have had little effect, because as the reviewer mentions the C/N of the reactive, likely N-rich, OM degrading during short-term core incubations will be lower than the bulk C/N.

However, the reviewer's comment did highlight for us that our approach did not account for anaerobic N loss which can be substantial in estuarine sediment (e.g. Seitzinger, 1988) and would cause a decoupling between DIN production from OM degradation in the sediment and the (comparatively low) DIN efflux. Using the suppressed DIN efflux and Redfield then underestimates the OM-derived DIC efflux, and consequently overestimates the DIC flux associated with e.g. $CaCO_3$ dissolution. In light of the importance of the DIC flux in evaluating the difference in OM degradation rates under submarine and subaerial conditions, we now include additional data, i.e. (1) the measured total alkalinity (TA) flux and (2) the $SO_4^{2-}$ reduction rate as TA source as calculated from the porewater $SO_4^{2-}$ gradient in the uppermost sediment. We arrive at a more robust estimation of the OM-derived DIC flux as an indicator of the OM degradation rate, mentioned in L415−418 in the revised manuscript and detailed in the SI. This approach and inclusion of the TA data also address a later comment by Reviewer #2 about $CaCO_3$ dissolution rates. Our new approach reduces the contribution of $CaCO_3$ dissolution (which is likely low in organic-rich coastal sediment; Krumins et al., 2013) and decreases the boost in OM degradation rates between in-situ and subaerial OM degradation rates (from 3–7 to 2–3; incorporated in L33, L630, 712, 743), but not to an extent that affects the overall conclusions of the work.

Minor comments:

Line 81-82: No need to highlight CH4 specifically. The point is valid for the balance between any given set of remineralization pathways.

Revised as detailed in the response to Reviewer #1's comment about [*Salinity and CH₄*].

Line 84-87: Consider rewording the sentence ("estuaries" appears to be used in both a general and a specific way in the same sentence) and add the reference.

Changed (L87−88).

Line 88: Should be "penetration depth".

Line 99: Make sure to use brackets correctly.

Line 123: Should be "dredged sediment".

Line 137-139: Should be "bottle incubations"

Line 139: Should be "Our results show..." or "We show..."

Line 141: Should be "the properties of OM influence..."

Line 213: Remove "an"

Line 383: Should be "overlying water".

Line 438: Should be "thus" not "this"

Line 478: Should be "may explain the observation that".

Line 607: Check the formatting of the citation.

Line 620: Should be "10 times faster than..."

Changed. We checked and corrected spelling and grammar in the revised manuscript.

Line 162: Use "end-members" or similar, rather than "realm"
Changed (L169).

Line 220-229: It would be useful for a reader trying to reproduce this method to know how the HF was disposed of during the protocol. Is it evaporated at some stage? This is a dangerous chemical and lab protocols need to be carefully designed.
We added the disposal of HF in the revised manuscript (L234, 237).

Line 229: Clarify that "MOM" here refers to the residue of the previous steps.
Changed (L239−240).

Line 283: It is not clear what is meant by "60% water-filled volume". It is difficult to estimate the porosity of freeze-dried and homogenized sediment after re-wetting due to changes in grain size distribution during these processing steps.
We added an assumption in the revised manuscript that porosity remained the same after sediment rewetting (L306−307).

Line 345: "Asterisk/stars" not "asteroids". Also unclear what the arrows in the plot indicate, should they be pointed in the opposite direction to link the numerical labels to the 2D fields in the x-y plot?
Corrected in Fig. 2 and its caption.

Line 245: Should be Table S3, not S2.
Corrected (L376).

Fig. 6: Please add the information about "marine" vs. "riverine" stations to the plot or legend.
Added. See Fig. 6.

Fig. 7: What do the stipple vs. solid lines indicate? Also show which sites are "marine" and "riverine".
In the revised manuscript, solid lines were used for all subplots, and site information was also added in Fig. 7.

Line 451: This is a valid consideration rather than a confounding factor.
Changed (L485).

Line 517: It is not unusual that soil OM contains similar chemical signatures to the higher plants that grow and decay in them. It is perhaps misleading to state that the correlation between BIT and the EMMM results implies low contribution of vegetation input, rather it should be stated that, as you conclude later, the plant signals are transferred via the soils (where they also pick up the BIT signal).

Because terrestrial OM pools are indeed not easily differentiated, we simplified the paragraph on proxy comparison (Fig. 8) (L546-554).

Line 551-552: The comparison with ocean margin trends is not useful here as the reactivity gradients in near-shore areas is more likely to be controlled by terrestrial inputs than by water-depth fractionation of sinking OM.
Based on the various reviews, we revised section 4.2, including removing the reference to pelagic water-depth trends, see L577 ff.

Line 564: Remove "ancient". There is no information about age of the material so this is not relevant.
Removed.

Line 567-568: Interpretation of carbonate dissolution needs to be expanded. Is there evidence for this in the data or literature?
As detailed above (see Reviewer #2's comment about [*Calculation OM-derived DIC fluxes*]), we have modified the calculation of the OM-derived DIC flux which also affected the estimation of DIC from $CaCO_3$ dissolution.

Line 577: The higher CH4 flux in a whole core incubation at a more freshwater site can be due to the higher position in the sediment column of the SMTZ. See e.g. the diagram in Fig. 7 of https://esd.copernicus.org/articles/13/633/2022/esd-13-633-2022.html.
We incorporated the SMTZ-shift into the text (L605).

Line 633: Also refer to Keil et al. (1994).
Added (L671).

Line 677-678: Separate the statements about global AOM estimates and dredging impacts, these are not obviously connected.
This section has been revised based on this and other reviewer comments and statements have been separated (L722-734).
* * *
**Reviewer #3:**
The manuscript of G. Wu et al. investigates the increase of organic matter (OM) degradation produced by anthropogenic perturbations such as dredging activity. The method stands on the elemental (C/N) and isotopic ($\delta$13C) characterization of organic matter of 49 sites from the Port of Rotterdam. Molecular characterization was done on 13 of the 49 sites, reoxydation experiments on 6 sites and in situ OM remineralization have been estimated on 2 sites. The authors conclude from their studies that Cl#1 - CN proxies are robust to identify contribution of marine versus non-marine OM sources, Cl#2 - marine

OM is more labile than terrestrial OM and CI#3-aerobic conditions accelerates OM remineralization compared to natural (mainly anaerobic) conditions.

From my point of view, this manuscript is not reaching the quality requirement to get published in BG: the rare new knowledges brought by this manuscript are only site specific. In the conclusion, the authors underline more general concepts (CI#1, CI#2, CI#3), however it corresponds to already well-established concept in biogeochemistry that does not present significant novelty.

We thank the reviewer for their time and effort in evaluating our manuscript. Reviewer #3's assessment about the quality and significance of the work strongly contrasts with evaluations by Reviewer #1 and #2, who highlighted the importance and high quality of the work. Uniquely, our work combines detailed sediment and OM characterization with incubation of intact and disturbed sediments to link $CO_2$ emission potential to OM properties (rather than content) and biogeochemical processes as function of environmental conditions. Whereas some of the broader concepts discussed (e.g. selective preservation, aerobic mineralization) are indeed established in biogeochemistry, the contribution of this study lies in applying and contextualizing these concepts within the complex settings of a heavily perturbed estuarine system, which is understudied in existing literature. In addition, the findings provide crucial insights into how human-induced disturbances interact with natural OM processing, with implications for carbon cycling. Such knowledge is also essential for informed policy making on processing dredging waste. As estuarine systems face increasing pressure of human perturbation, we believe this study represents a meaningful contribution to the field, as also noted by both Reviewer #1 and #2.

[*$O_2$ fluxes*] In addition, a detailed reading of the manuscript reveal that the scientific hypothesis defended along the manuscript (that perturbation increases source-dependent organic matter degradation rates in estuarine sediment) is not properly argued. While the source characterization of the OM received large analytical efforts, and a valid interpretation supported by some modelisation, four others critical step of the demonstration seems not strong enough. First, and most important, the demonstration of a source and composition dependence of OM degradation rate is not convincing. The authors build their hypothesis from the literature and a reasonable intuition but that is in complete opposition with the dataset presented. Indeed, the two sites, selected because they present very contrasted OM origin and composition, present very similar O2 fluxes that is interpreted -reasonably- as a similar intensity of remineralization rate. This result should better be used to underline a limit of the observations from the literature. Instead, the authors use an artefact (normalization by the TOC) to build a contrast between the two stations while the relation between quantity of OM and remineralization rate have not been presented. It seems more rational to conclude from their dataset that OM quantity does not play an important role in deposit, probably because it is in large excess.

We include further discussion about the similarity in short-term $O_2$ fluxes and the underlying reasons (similar processes acting at the sediment-water interface) in the revised manuscript (specifically L591–592. We further agree that sedimentary organic carbon turnover in the core incubation experiments might to some extent depend on $O_2$ availability (L593–594). We have kept the TOC-normalized rates in

the manuscript; this approach is commonly used to assess the relative reactivity of OM (e.g. Freitas et al., 2025; Zander et al., 2020, 2022), especially when TOC content differs considerably between sites (in this case 2.2 wt.% and 5 wt.%). The TOC-normalization allows comparing OM degradation rates between sites with different OM compositions (more fresh algal material at site 115 compared to the more refractory terrestrial-derived OM at site 21A) and, furthermore, between different incubation conditions (Fig. 5d and 6b).

[*Algae-derived organic matter*] Second in importance, there is no consideration on primary production, while freshly produced algea are probably more labile than any other carbon source. In other word the local recycling of carbon between photosynthesis and respiration is neglected in this manuscript while it is probably very important in this very shallow environment.

Algal material and its degradation plays a central role in the manuscript. We now explicitly address the role of freshly produced algae in controlling short-term O2 fluxes (L591−592). Study of benthic primary productivity as $O_2$ source was not part of this study, but likely insignificant in these highly turbid waters.

[*Site selection and methodology*] Third, while the introduction suggests a study covering many sites and embracing some spatial variability, the comparison between perturbated and not perturbated sediment is done only on 2 sediment, which seems not enough to support conclusions that could be generalized to others estuaries, and thus limits the interest of the results.

We have detailed the rationale for site selection for the various analyses and experiments, please see our response to the editor's comment about [*Site selection and methodology*]. We note that this study incorporates a range of analyses and experiments for integrated insight into OM properties and degradation rates; within the scope of the project, it was only feasible to perform whole-core incubations with sediments from two contrasting and important dredging sites.

The open-air bottle experiments based on six sediments further supported the observed relationship from whole-core incubation: marine sediment OM was more reactive/biodegradable than riverine sediment OM. Therefore, we believe that our work does reflect a general relationship between OM composition and $CO_2$ release potential in human-impacted estuaries (e.g. the Elbe estuary; Zander et al., 2022), but the spatial distribution of the more degradable OM (e.g. algal material) may differ between systems, now also mentioned in the Implications (L695−700)

[*Subaerial bottle incubation experiment*] Fourth, the methodology of perturbation does not seem adapted to an estuarine environment, since it corresponds to a not water-saturated sediment more adapted to mimic soil aeration than resuspension induced by dredging activities.

Please see our response to the editor's comment about experimental approach for the [*Subaerial bottle incubation experiment*].

[*Sample processing*] Additionally, the dry freezing of the sediment before incubation would certainly modify the properties of the reactive organic matter – which have not been tested.

Please see our response to the editor's comment about [*Sample processing*] above.

**References**

Freitas, N. L., Walter Anthony, K., Lenz, J., Porras, R. C., and Torn, M. S.: Substantial and overlooked greenhouse gas emissions from deep Arctic lake sediment, Nat Geosci, https://doi.org/10.1038/s41561-024-01614-y, 2025.

Krumins, V., Gehlen, M., Arndt, S., Van Cappellen, P., and Regnier, P.: Dissolved inorganic carbon and alkalinity fluxes from coastal marine sediments: Model estimates for different shelf environments and sensitivity to global change, Biogeosciences, 10, 371–398, https://doi.org/10.5194/bg-10-371-2013, 2013.

Seitzinger, S. P.: Denitrification in freshwater and coastal marine ecosystems: Ecological and geochemical significance, Limnol Oceanogr, 33, 702–724, https://doi.org/10.4319/lo.1988.33.4part2.0702, 1988.

Zander, F., Heimovaara, T., and Gebert, J.: Spatial variability of organic matter degradability in tidal Elbe sediments, J Soils Sediments, 20, 2573–2587, https://doi.org/10.1007/s11368-020-02569-4, 2020.

Zander, F., Groengroeft, A., Eschenbach, A., Heimovaara, T. J., and Gebert, J.: Organic matter pools in sediments of the tidal Elbe river, Limnologica, 96, 125997, https://doi.org/10.1016/j.limno.2022.125997, 2022.

---

## Referee Report (RR1)

Dear editors,

This manuscript investigates how organic matter (OM) sources and human-induced perturbations influence OM degradation in the highly dynamic and heavily dredged estuarine system of the Port of Rotterdam. The authors address an important and timely question about shifts in OM provenance and oxygen exposure, especially through dredging and subaerial sediment handling, and consequences on carbon mineralization and potential $CO_2$/$CH_4$ emissions. The study is well motivated given the disproportionate role of estuaries in the global carbon cycle and the current uncertainty associated with their response to anthropogenic disturbance.

The authors present a comprehensive dataset. Themulti-proxy approach is a strong asset and clearly demonstrates spatial variability in OM sources along the salinity gradient, as well as consistent differences in reactivity between marine (more labile) and riverine/terrestrial (more refractory) OM. A key contribution of the manuscript is the demonstration that perturbation and associated oxygen exposure substantially enhance OM degradation rates, by a factor of two to three, highlighting a carbon-cycle impact of dredging that is often overlooked.

The manuscript is clearly written, the methods are rigorous, and the results support the conclusions. Some aspects could be clarified further, such as the limitations of using freeze-dried sediments for subaerial incubations or the quantitative implications for carbon budgets at system scale.

Overall, this is a valuable and well-constructed study that provides meaningful insights into OM dynamics in disturbed estuarine environments. I therefore support publication in its current form.

---

## Author Response (AR2)

Dear Editor and Reviewers,

We appreciate your positive recommendation as well as the insightful comments on our manuscript titled '*Perturbation increases source-dependent organic matter degradation rates in estuarine sediments*". Below, we respond in detail to the minor comments raised by Reviewer #2. For each point, we describe the corresponding revisions made and indicate how these comments have been addressed in the revised manuscript.

Kind regards,
Guangnan Wu, on behalf of the coauthors

NB: Our response is given in red below, also referring to the line numbers in the revised manuscript with Word's **simple markup** where the edited text can be found.
* * *
**Reviewer #2:**

Overall the authors have addressed the comments from the discussion round in a thorough and integrated way. I support their assertion that it is valid to normalize mineralization rates to carbon content and that the results support a genuine difference in reactivity between marine and terrestrial sediment OM. The issue I raised concerning the use of Redfield stoichiometry to estimate DIC fluxes from ammonium data in the whole-core incubation has been circumnavigated in the new version through a direct correction of the DIC data to remove the contribution of $CaCO_3$ dissolution. I am satisfied that this approach is valid.

We thank the reviewer their positive assessment of the revised manuscript.

However, I suggest some further clarification of how the results of this section are reported and discussed. The authors highlight that DIC production is approx. 4-5 times higher on a molar basis than $O_2$ consumption. Later in the discussion (Line 625-627) there is a statement that aerobic mineralization accounts for only 25-30% of total mineralization, and therefore that anaerobic mineralization is important. This is presumably a reference to the excess DIC production assuming standard stoichiometry of degrading OM, but it is not made clearly. I would suggest to add already in the Methods section the approach that will be used to differentiate aerobic from anaerobic mineralization rates when processing the data.

We have now clarified in the Methods section how aerobic and anaerobic mineralization rates were differentiated (Line 294–297).

Minor comments:
Line 139 "This study show"
Line 156 "being subaerially in a holding basin"
Line 298 "while avoiding oxygen supply as a limitation" (can be removed)

Line 584-587 sentence is too long

Changed. We also checked and corrected spelling and grammar in the revised manuscript.